

# Novel hybrid firefly algorithm: an application to enhance XGBoost tuning for intrusion detection classification

Miodrag Zivkovic[1], Milan Tair[1], Venkatachalam K[2], Nebojsa Bacanin[1], Štěpán Hubálovský[2] and Pavel Trojovský[3]

[1] Singidunum University, Belgrade, Serbia
[2] Department of Applied Cybernetics, Faculty of Science, University of Hradec Králové, Hradec Kralove, Hradec Kralove, Czech Republic
[3] Department of Mathematics, Faculty of Science, University of Hradec Králové, Hradec Kralove, Hradec Kralove, Czech Republic

## ABSTRACT

The research proposed in this article presents a novel improved version of the widely adopted firefly algorithm and its application for tuning and optimising XGBoost classifier hyper-parameters for network intrusion detection. One of the greatest issues in the domain of network intrusion detection systems are relatively high false positives and false negatives rates. In the proposed study, by using XGBoost classifier optimised with improved firefly algorithm, this challenge is addressed. Based on the established practice from the modern literature, the proposed improved firefly algorithm was first validated on 28 well-known CEC2013 benchmark instances a comparative analysis with the original firefly algorithm and other state-of-the-art metaheuristics was conducted. Afterwards, the devised method was adopted and tested for XGBoost hyper-parameters optimisation and the tuned classifier was tested on the widely used benchmarking NSL-KDD dataset and more recent USNW-NB15 dataset for network intrusion detection. Obtained experimental results prove that the proposed metaheuristics has significant potential in tackling machine learning hyper-parameters optimisation challenge and that it can be used for improving classification accuracy and average precision of network intrusion detection systems.

# INTRODUCTION

The firefly algorithm (FA), proposed by *Yang (2009)*, is a swarm intelligence algorithm designed for exploration, exploitation, and local search of solutions, inspired by social behaviour and flashing activities exhibited by the fireflies. The original FA algorithm is tested against the updated CEC2013 benchmark function set in this article. Also, this article presents the performance of a well-known XGBoost classifier, whose parameters have been optimised using the FA algorithm for the problem of Network Intrusion Detection (NIDS) optimisation. Different NIDS have a simple purpose: to monitor network traffic and detect malicious user activities. They are usually implemented as nodes on strategic points in the network.

Corresponding author
Nebojsa Bacanin,
nbacanin@singidunum.ac.rs

According to *Hodo et al. (2016)*, NIDS are reliable when dealing with outside threats but are inefficient for determining the extent of damage from an attack. Core issues with these systems are rates of false positives (FP) and false negatives (FN). FPs occur when the system wrongly classifies regular activities as malicious, and FNs occur when the system fails to properly classify malicious activities as such. Different approaches are used to attempt to solve these issues. Machine learning (ML) algorithms present one possible type of solution. ML solutions attempt to find optimal hyper-parameters to optimise classifiers for different network activity datasets to increase detection efficiency.

Among ML algorithms, there are different approaches for optimising the ML model (*Ikram et al., 2021*; *Thaseen & Kumar, 2017*). Swarm intelligence (SI) algorithms usually have good performance (*Bergstra et al., 2011*; *Jiang et al., 2020*; *Bacanin et al., 2020*). This study tests the performance of a well-known XGBoost classifier for classifying NIDS events from the NSL-KDD (Network Security Laboratory - Knowledge Discovery and Data Mining) dataset (*Dhanabal & Shantharajah, 2015*), which are, according to *Protić (2018)*, effective in evaluating intrusion detection systems. The second experiment test the performance of the XGBoost classifier on a more recent NIDS UNSW-NB15 dataset (*Moustafa & Slay, 2015*). The hyper-parameters of the classifier are optimised using a proposed improved FA algorithm.

The motivation behind the approach suggested in this research was to enhance the basic implementation of FA further and improve the classification capabilities of XGBoost classifier. According to the no free lunch theorem, an universal optimisation algorithm that can solve all optimisation problems does not exist. Additionally, it is always possible to improve the existing optimisation algorithms. In this context, the research proposed in this paper is focused on improving the solving of the very important challenge in intrusion detection systems by using the XGBoost classifier, and in order to do so, a new, improved FA algorithm has been developed. The most important contributions of this paper are three-fold:

• A novel enhanced FA metaheuristics has been developed by specifically targeting the well-known deficiencies of the original FA implementation;

• The developed method was later used to help establish the proper hyper-parameters values and improve the XGBoost classifier accuracy for the intrusion detection classification problem;

• The proposed method results were compared with other notable swarm intelligence algorithms, which were further investigated for the XGBoost optimisation problem.

The rest of the paper is structured as follows. The following section introduces the problem of optimisation and various types of optimisation algorithms, focusing on related work in machine learning algorithms. The paper presents the proposed model, describes the set of CEC2013 functions, the firefly algorithm's performance assessed by this benchmark, gives an overview of the experimental setup for the second part of the paper. Results, comparative analysis and result discussion section, follow the materials and methods section. Finally, the conclusion of this paper is given with suggested future work propositions.

## BACKGROUND

This section introduces NIDS, the problem of optimisation, in general, and concerning NIDS, and different algorithmic approaches to optimising NIDS network event classification methods. Different machine learning approaches are presented towards the end of the section, leading up to the overview of works related to this problem.

### The problem of network intrusion detection

In the last two decades, the web has become the centre stage for many businesses, social, political and other activities and transactions that all happen on the global network. Endpoints of those network transactions are users, usually located within smaller computer networks, such as companies, small Internet provider sub-networks etc. Therefore, security has become an important issue for the contemporary Internet user, even though different intrusion detection solutions have been around for almost 40 years (*Neupane, Haddad & Chen, 2018*). There are many solutions created to protect users from malicious activities and attacks (*Patel, Qassim & Wills, 2010*; *Mugunthan, 2019*).

According to *Neupane, Haddad & Chen (2018)*, traditional NIDS come in the forms of firewalls, and statistical detection approaches usually applied either on the transport or the application layers, which require extensive setup, policy configurations etc. More modern systems use sophisticated approaches. According to *Sathesh (2019)*, ML is used to solve the problem of intrusion detection, even though these approaches have different challenges, as reported by *Jordan & Mitchell (2015)*. Regardless, ML approaches are efficient in finding optimal solutions for time-consuming problems, such as training efficient NIDS network event classifiers. Different solutions are based on various types of ML methods, such as artificial neural networks (ANN), evolutionary algorithms (EA), and other supervised and unsupervised learning methods, according to *Verwoerd & Hunt (2002)*.

When creating and evaluating a NIDS, it is important to measure its performance accurately. For this reason, previously mentioned false positives (FP) and false negative (FN) measurements are used together with true positive (TP) and true negative (TN) measurements to correctly evaluate the classification accuracy of a NIDS, according to the general formula shown in Eq. (1).

$$ACC = (TP + TN)/(TP + FP + TN + FN) \tag{1}$$

From values TP, TN, FP and FN, it is possible to also determine the system's sensitivity, specificity, fallout, miss rate, and prevision through methods presented in Eqs. (2)–(6):

$$Sensitivity = TP/(TP + FN) \tag{2}$$
$$Specificity = TN/(TN + FP) \tag{3}$$
$$Fallout = FP/(TN + FP) \tag{4}$$
$$Missrate = FN/(TP + FN) \tag{5}$$
$$Precision = TP/(TP + FP) \tag{6}$$

## Optimisation and optimisation algorithms

Optimisation aims to find an optimal or near-optimal solution for a certain problem within the given set of constraints. Many population-based stochastic meta-heuristics were developed for solving the problem of optimisation, according to *Beheshti & Shamsuddin (2013)*.

Non-deterministic polynomial-time-hard problems are hard to solve with traditional deterministic algorithms. They can take a long time to complete on commonly available hardware. Therefore, these solutions are usually impractical.

On the other hand, optimal solutions to these types of problems can be found using stochastic meta-heuristics, which do not guarantee an optimal solution, but acceptable sub-optimal ones in reasonable time-frames, according to *Spall (2011)*. Commonly, these algorithms are labelled as Machine Learning Algorithms (MLA).

## Swarm intelligence algorithms

A special type of nature-inspired stochastic meta-heuristic MLA are population-based algorithms, among which are swarm intelligence algorithms (SIA). These algorithms inspire different naturally occurring systems, where individual self-organising agents interact with each other and their environment without a centralised governing component. These systems give an impression of globally coordinated behaviour and have inexpensive abilities in solving very demanding optimisation problems (*Mavrovouniotis, Li & Yang, 2017*).

The most notable and popular methods that have proven themselves as powerful optimiser with respectable performances include the ant colony optimisation (ACO) introduced by *Dorigo, Birattari & Stutzle (2006)*, artificial bee colony (ABC) proposed by *Karaboga & Basturk (2007)*, particle swarm optimisation (PSO) developed by *Kennedy & Eberhart (1995)*, as well as the FA, introduced by *Yang (2009)* and used as a foundation for the algorithm proposed in this paper. More recent algorithms that have shown good results include the grey wolf optimiser (GWO) (*Mirjalili, Mirjalili & Lewis, 2014*), moth search (MS) (*Wang, 2018*), monarch butterfly algorithm (MBA) (*Wang, Deb & Cui, 2019*), whale optimisation algorithm (WOA) (*Mirjalili & Lewis, 2016*), and the Harris hawk's optimisation (HHO) (*Heidari et al., 2019*). Additionally, the differential evolution algorithm (*Karaboğa & Okdem, 2004*) and the co-variance matrix adaptation (*Igel, Hansen & Roth, 2007*) approaches have also recently exhibited outstanding performances. Recently, algorithms inspired by the properties of the mathematical functions gained popularity among scientific circles, and the most notable algorithm is the sine-cosine algorithm (SCA), which was proposed by *Mirjalili (2016)*. SCA was also utilised in this research to hybridise the basic FA search.

The application of the metaheuristics discussed take on a wide spectrum of different problems with NP-hardness in the information technologies field. Some of the such applications are with the problem of global numerical optimisation (*Bezdan et al., 2021b*), scheduling of tasks in the cloud reliant systems (*Bezdan et al., 2020b; Bacanin et al., 2019a; Zivkovic et al., 2021b*), the problems of wireless sensors networks such as localisation of nodes and the network lifetime prolonging (*Zivkovic et al., 2020a; Bacanin*

*et al., 2019b*; *Zivkovic et al., 2020b*; *Bacanin et al., 2022b*; *Zivkovic et al., 2021d*), artificial neural networks optimisation (*Strumberger et al., 2019*; *Milosevic et al., 2021*; *Bezdan et al., 2021c*; *Cuk et al., 2021*; *Stoean, 2018*; *Bacanin et al., 2022c*; *Gajic et al., 2021*; *Bacanin et al., 2022a*; *Jnr, Ziggah & Relvas, 2021*; *Bacanin et al., 2021a*, *2022d*, *2021b*, *2021c*), histological slides or MRI classifier optimisation (*Bezdan et al., 2020a*, *2021a*, *Lichtblau & Stoean, 2019*; *Postavaru et al., 2017*; *Basha et al., 2021*), and last but not least the COVID-19 case prediction (*Zivkovic et al., 2021a*, *2021c*).

These algorithms, on their own, have strengths and weaknesses when applied to different problems, and often, they are used to optimise different higher-level models and their hyper-parameters instead of being used to perform classification on their own. This article presents this synthesis of an optimisation algorithm used for hyper-parameter tuning and optimising a higher-order classification system.

### Related work

Applications of ML algorithms have been reported in many scientific and practical fields in the industry, as well as for NIDS, as reported by *Tama & Lim (2021)* and *Ahmed & Hamad (2021)*. Specifically for the problem of NIDS optimisation, solutions exist that are based on particle swarm optimisation (PSO) (*Jiang et al., 2020*), artificial neural networks (ANN) and support vector machines (SVM) (*Mukkamala, Janoski & Sung, 2002*), naive Bayesian (NB) (*Mukherjee & Sharma, 2012*), K-nearest neighbour (KNN) (*Govindarajan & Chandrasekaran, 2009*), and in combination with other classifiers, as presented by *Dhaliwal, Nahid & Abbas (2018)*, *Ajdani & Ghaffary (2021)* and *Bhati et al. (2021)*.

## METHODS

The original implementation of the FA is shown in this section, followed by the descriptions of known and observed flaws of the original FA. The section suggests improvements to the original algorithm to address the described flaws.

### The original version of the firefly algorithm

*Yang (2009)* has suggested a swarm intelligence system that was inspired by the fireflies' lighting phenomenon and social behaviour. Because the behaviour of actual fireflies is complicated, the FA metaheuristics model, with certain approximations, was proposed.

The fitness functions are modelled using the firefly's brightness and attraction. In most FA implementations, attractiveness depends on the brightness, determined by the objective function's value. In the case of minimisation problems, it is written as *Yang (2009)*:

$$I(x) = \begin{cases} 1/f(x), & \text{if } f(x) > 0 \\ 1 + |f(x)|, & \text{if } f(x) \leq 0 \end{cases} \qquad (7)$$

where $I(x)$ is the attractiveness and $f(x)$ represents the objective function's value at $x$, which is the location.

Therefore, the attractiveness of a firefly is indirectly proportional to the distance from the source of light (*Yang, 2009*):

$$I(r) = \frac{I_0}{1 + \gamma \times r^2} \tag{8}$$

When modelling systems where the environment absorbs the light, the FA uses the light absorption coefficient parameter $\gamma$. $I(r)$ and $I_0$ are the intensities of the light at the distance of $r$ and the source. Most FA implementations combine effects of the inverse square law for distance and $\gamma$ to approximate the following Gaussian form *Yang (2009)*:

$$I(r) = I_0 \cdot e^{-\gamma \times r^2} \tag{9}$$

As indicated in Eq. (10), each firefly employs $\beta$ (representing attractiveness), which is proportionate to the intensity of the firefly's light, which is reliant on distance.

$$\beta(r) = \beta_0 \cdot e^{-\gamma \times r^2} \tag{10}$$

where $\beta_0$ represents the attractiveness at $r = 0$. However, Eq. (10) is commonly swapped for Eq. (11) (*Yang, 2009*):

$$\beta(r) = \beta_0 / \left(1 + \gamma \times r^2\right) \tag{11}$$

Based on Eq. (11), the equation for a random firefly $i$, moving in iteration $t + 1$ to a new location $x_i$ in the direction of another firefly $j$, which has a greater fitness value, according to the original FA, is (*Yang, 2009*):

$$x_i^{t+1} = x_i^t + \beta_0 \cdot e^{-\gamma \times r_{i,j}^2}(x_j^t - x_i^t) + \alpha^t(\kappa - 0.5) \tag{12}$$

where $\alpha$ is the randomisation parameter, $\kappa$ is the random uniformly distributed number, and $r_{i,j}$ is the distance between fireflies $i$ and $j$. Values that often give good results for most problems for $\beta_0$ and $\alpha$ are 1 and [0, 1]. The $r_{i,j}$ is calculated as follows, and represents the Cartesian distance:

$$r_{i,j} = ||x_i - x_j|| = \sqrt{\sum_{k=1}^{D} \left(x_{i,k} - x_{j,k}\right)^2} \tag{13}$$

where $D$ is the number of parameters of a specific problem.

## Reasons for improvements

The original FA has performed exceptionally for many benchmarks (*Yang & He, 2013*) and practical problems (*Strumberger et al., 2019*). Past research suggests that the original FA has several flaws regarding exploration and an inappropriate intensification-diversification balance (*Strumberger, Bacanin & Tuba, 2017*; *Xu, Zhang & Lai, 2021*; *Bacanin & Tuba, 2014*). The lack of diversity is noticeable in early iterations, when the algorithm cannot converge to optimum search space areas in certain runs, resulting in low mean values. In such cases, the original FA search technique (Eq. (12)), which mostly performs exploitation, is incapable of directing the search to optimal domains.

In contrast, the FA achieves satisfactory results when random solutions are created randomly in optimal or near-optimal areas during the initialisation phase.

An examination of the original FA search equation (Eq. (12)) reveals that it lacks an explicit exploration technique. Some FA implementations employ the dynamic randomisation parameter $\alpha$, which is continuously reduced from its starting value $\alpha$ to the specified threshold $\alpha_{min}$, as shown in Eq. (14). As a result, at the start of a run, exploration is prioritised, whereas subsequent iterations shift the balance between intensity and diversification toward exploitation (*Wang et al., 2017*). However, based on simulations, it is concluded that the use of dynamic $\alpha$ is insufficient to improve FA exploration skills, and the suggested technique only somewhat alleviates this problem.

$$\alpha^{t+1} = \alpha^t \cdot (1 - t/T), \tag{14}$$

where $t$ and $t + 1$ are current and next iterations, and $T$ is the maximum iteration count in a single run.

Past research has shown that FA exploitation abilities are effective in addressing a variety of tasks, and FA is characterised as a metaheuristic with substantial exploitation capabilities (*Strumberger, Bacanin & Tuba, 2017*; *Xu, Zhang & Lai, 2021*; *Bacanin & Tuba, 2014*).

## Novel FA metaheuristics

This work proposes an improved FA that tackles the original FA's flaws by using the following procedures:

- A technique for explicit exploration based on the exhaustiveness of the answer;
- gBest chaotic local search (CLS) approach.
- Hybridisation with SCA search by doing either FA or SCA search at random in each cycle based on a produced pseudo-random value.

The FA's intensification may be improved further by applying the CLS mechanism, as demonstrated in the empirical portion of this work. A novel FA is dubbed chaotic FA with improved exploration due to proposed modifications (CFAEE-SCA).

### *Explicit exploration mechanism*

The purpose of this mechanism is to ensure the convergence to the best section of the search space early on, while facilitating exploration around the parameter bounds of the current best individual $x^*$ later on. Each solution is represented using an additional attribute *trial*. It increases this attribute when it cannot further improve the solution with the original FA search (Eq. (12)). When the *trial* parameter reaches a set *limit*, the individual is swapped for a random one picked from the search space in the same way as in the setup phase:

$$x_{i,j} = l_j + (u_j - l_j) \cdot rand, \tag{15}$$

where $x_{i,j}$ is the $j$-th component of $i$-th individual, $u_j$ and $l_j$ are the upper and lower search boundaries of the $j$-th parameter, and *rand* is a random number in range [0, 1], from a uniform distribution.

A complete solution is one for which *trial* exceeds the *limit*. This term was adapted from the well-known ABC metaheuristics (*Karaboga & Basturk, 2008*), which have efficient exploration mechanisms (*Moradi et al., 2018*).

When the algorithm fails to find appropriate areas of the search space, replacing the exhausted solution with a pseudo-random person improves search performance early on. Later on, this type of substitution wastes functions evaluations. As a result, in subsequent iterations, the random replacement technique is replaced by the directed replacement mechanism around the bottom and higher parameter values of the population's solutions:

$$x_{i,j} = Pl_j + (Pu_j - Pl_j) \cdot rand, \tag{16}$$

where $Pl_j$ and $Pu_j$ are the lowest and highest values of the $j$-th component from the whole population $P$.

### The gBest CLS strategy

Chaos is responsive to the initial conditions of non-linear and deterministic systems (*Alatas, 2010*). Chaotic search is more efficient than the ergodic (*dos Santos Coelho & Mariani, 2008*) because many sequences can be created by modifying the initial values.

Literature reports many chaotic maps. After testing, it was determined that the logistic map yields the most favourable results in the case of the suggested innovative FA. The logistic map has been used in a variety of swarm intelligence methodologies so far (*Li et al., 2012*; *Chen et al., 2019*; *Liang et al., 2020*). The logistic map used by the proposed method is defined in $K$ steps as:

$$\sigma_{i,j}^{k+1} = \mu\sigma_{i,j}^k(1 - \sigma_{i,j}), \; k = 1, 2, \ldots K, \tag{17}$$

where $\sigma_{i,j}^k$ and $\sigma_{i,j}^{k+1}$ are chaotic variable for the $i$-th solution's $j$-th component in steps $k$ and $k + 1$, and $\mu$ is a control variable. $\sigma_{i,j} \neq 0.25, 0.5$ and $0.75$, $\sigma_{i,j} \in (0, 1)$ and $\mu$ is set to 4. This value was determined empirically by *Liang et al. (2020)*.

The proposed method integrates the global best (gBest) CLS strategy. The chaotic search is performed around the $x^*$ solution. Equations (18) and (19) show how a new $x^*$ ($x'^*$) is created in each step $k$, for component $j$ of $x^*$:

$$x_j'^* = (1 - \lambda)x_j^* + \lambda S_j \tag{18}$$

$$S_j = l_j + \sigma_j^k(u_j - l_j) \tag{19}$$

where Eq. (17) determines $\sigma_j^k$, and $\lambda$ is the dynamic shrinkage parameter dependant on *FFE* (current fitness function evaluation) and *maxFFE* (maximum number of fitness function evaluations):

$$\lambda = (maxFFE - FFE + 1)/maxFFE \tag{20}$$

Better exploitation-to-exploration equilibrium is formed around the $x^*$ by employing dynamic *lambda*. Earlier in the execution, a larger search radius around the $x^*$ was performed, whereas later, a fine-tuned exploitation commenced.

When the maximum number of iterations is used as the termination condition, the *FFE* and *maxFFE* can be substituted with $t$ and $T$.

The CLS strategy is used to enhance $x^*$ in $K$ steps. If the $x'^*$ achieves greater fitness than the $x^*$, the CLS method is ended, and the $x^*$ is replaced with $x'^*$. However, if the $x^*$ could not be improved in $K$ stages, it is maintained in the population.

### SCA search

SCA proposes the use of the following updating Eq. (21) in both phases:

$$
\begin{aligned}
X_i^{t+1} &= X_i^t + r_1 \times sin(r_2) \times \left| r_3 P_i^t - X_i^t \right| \\
X_i^{t+1} &= X_i^t + r_1 \times cos(r_2) \times \left| r_3 P_i^t - X_{i_i}^t \right|
\end{aligned}
\tag{21}
$$

where $X$ is the position of the current solution in the $i$-th dimension after the $t$-th iteration, $P_i$ is the destination at the $i$-th dimension, and $r_1$, $r_2$ and $r_3$ are random numbers.

A combination of Eq. (21) is show in Eq. (22):

$$
X_i^{t+1} = \begin{cases} X_i^{t+1} = X_i^t + r_1 \times sin(r_2) \times \left| r_3 P_i^t - X_i^t \right|, & r_4 < 0.5 \\ X_i^{t+1} = X_i^t + r_1 \times cos(r_2) \times \left| r_3 P_i^t - X_{i_i}^t \right|, & r_4 \geq 0.5 \end{cases}
\tag{22}
$$

where $r_4$ is a random value in range [0, 1].

The preceding equation demonstrates that the algorithm's four main parameters are $r_1$, $r_2$, $r_3$, and $r_4$. The $r_1$ parameter determines the region (movement direction) of the next location; it might be inside or outside the area between the destination and solution. The $r_2$ parameter specifies the amplitude and direction of the movement (towards the destination or outwards). The $r_3$ parameter assigns a random weight to the destination in order to reduce ($r_3 1$) or accentuate ($r_3 > 1$) the impacts of the destination in the distance definition. The $r_4$ parameter is used to alternate between sine and cosine components.

The SCA search algorithm is included in the proposed method in the following fashion. Each cycle generates a pseudo-random number. If the resulting value is more than 0.5, the FA search algorithm does. Otherwise, it executes the SCA search described in Eq. (22).

### Chaotic FA with enhanced exploration and SCA search pseudo-code

A few factors should be examined to efficiently include the exploration mechanism and gBest CLS approach into the original FA. First, as previously indicated, the random replacement method should be used in the early stages of execution, while the guided one would produce superior outcomes later on. Second, the gBest CLS technique would not produce substantial gains in early iterations since the $x^*$ would still not converge to the optimal area, wasting *FFEs*.

The extra control parameter $\psi$ is introduced to govern the behaviour as mentioned earlier. If $t < \psi$, the exhausted population solutions are replaced randomly Eq. (15) without activating the gBest CLS. Otherwise, it executes the guided replacement mechanism Eq. (16) and activates the gBest CLS.

The original FA search suggested approach uses dynamic *alpha* to fine-tune, according to Eq. (14). Based on the pseudo-random value, the method alternates between FA and SCA in each round.

---

**Algorithm 1  The CFAEE-SCA pseudo-code**

Initialise control parameters $N$ and $T$

Initialise search space parameters $D$, $u_j$ and $l_j$

Initialise CFAEE-SCA parameters $\gamma$, $\beta_0$, $\alpha_0$, $\alpha_{min}$, $K$ and $\phi$

Initialise random population $P_{init} = \{x_{i,j}\}$, $i = 1, 2, 3 \cdots, N; j = 1, 2, \cdots, D$ using Eq. (15) in the search space

**while** $t < T$ **do**

  **for** $i = 1$ to $N$ **do**

    **for** $z = 1$ to $i$ **do**

      **if** $I_z < I_i$ **then**

        Generate pseudo-random value $rnd$

        **If** $rnd > 0.5$ **then**

          Perform FA search

          Move solution $z$ in the direction of individual $i$ in $D$ dimensions (Eq. (12))

          Attractiveness changes with distance $r$ as $\exp[-\gamma r]$ (Eq. (10))

          Evaluate new solution, swap the worse individual for a better one and update light intensity

        **else**

          Perform SCA search

          Move solution $z$ in $D$ dimensions (Eq. (22))

        **end if**

      **end if**

    **end for**

  **end for**

  **it** $t < \phi$ **then**

    Swap solutions where $trial = limit$ for random ones using Eq. (15)

  **Else**

    Swap solutions where $trial = limit$ for others, using Eq. (16)

    **for** $k = 1$ to $K$ **do**

      Perform gBest CLS around the $x^*$ using Eqs. (17)–(19) and generate $x'^*$

      Retain better solution between $x^*$ and $x'^*$

    **end for**

  **end if**

  Update parameters $\alpha$ and $\lambda$ using Eqs. (14) and (20)

**end while**

Return the best individual $x^*$ from the population

Post-process results and perform visualisation

---

Taking everything above into account, Algorithm 1 summarises the pseudo-code of the proposed CFAEE-SCA.

The flowchart of the proposed CFAEE-SCA algorithm is given in the Fig. 1.

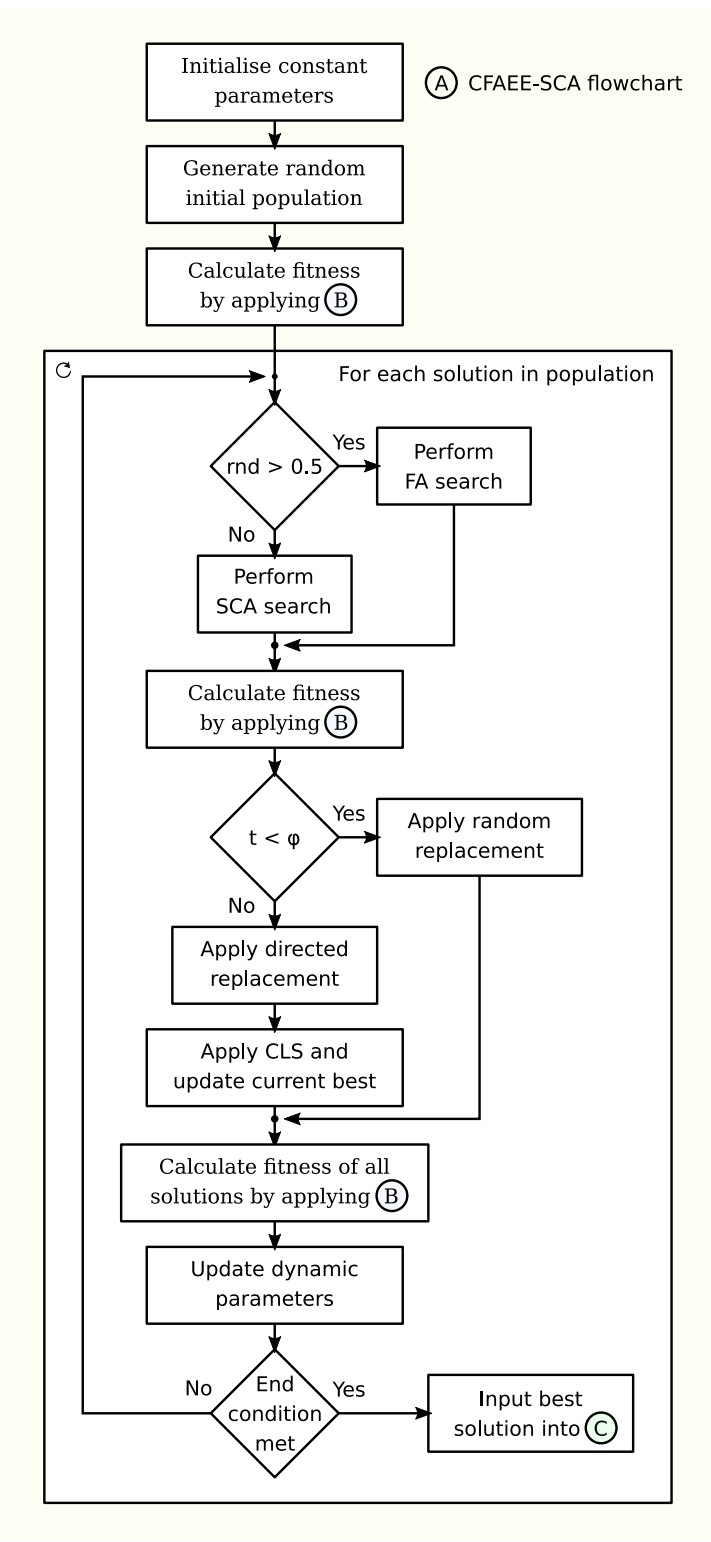

**Figure 1 Flowchart of the proposed CFAEE-SCA algorithm.**

### The CFAEE-SCA complexity and drawbacks

Because the most computationally costly portion of the swarm intelligence algorithm is the objective evaluation (*Yang & He, 2013*), the number of FFEs may be used to assess the complexity of the method.

The basic FA evaluates objective functions during the startup and solution update stages. When updating solutions, the FA utilises one main loop for $T$ iterations and two inner loops that go through $N$ solutions, according to the Eq. (12) (*Yang & He, 2013*).

Basic FA metaheuristics have a worst-case complexity of $O(N) + O(N^2 \cdot T)$, including the initialisation phase. However, if $N$ is large enough, one inner loop may be used to rate the beauty or brightness of all fireflies using sorting algorithms. Complexity in this situation is $O(N) + O(N \cdot T \cdot \log(N))$ (*Yang & He, 2013*).

Because of the explicit exploration mechanism and the gBest CLS method, the suggested CFAEE-SCA has a higher complexity than the original FA. In the worst-case situation, if $limit = 0$, all solutions will be replaced in every iteration, and if $\phi = 0$, the gBest CLS approach will be activated during the whole run. Assuming that $K$ is set to 4, the worst-case CFAEE-SCA complexity is stated as: $O(N) + O(T \cdot N^2) + O(T \cdot N) + O(4 \cdot T)$. In practice, however, the complexity is substantially lower due to $limit$ and $\psi$ control parameter modifications.

The CFAEE-SCA has certain drawbacks over the original design, including the use of new control parameters $limit$ and $\psi$. However, the values of these parameters may be easily determined by performing empirical simulations. Furthermore, as proven in the next sections, the CFAEE-SCA outperforms the original FA for benchmark tasks and the XGBoost optimisation challenge from the machine learning domain.

## RESULTS OF PROPOSED ALGORITHM AGAINST STANDARD CEC2013 BENCHMARK FUNCTION SET

The CEC2013 benchmark functions suite consists of 28 challenging benchmark function instances belonging to the different classes. Functions 1–5 belong to the group of uni-modal instances, functions 6–20 are multi-modal instances, while functions 21–28 belong to the composite functions family. The CEC2013 functions list is presented in Table 1. The challenge is to minimise the functions. Each class of functions has its purpose - uni-modal benchmarks test the exploitation, multi-modal benchmarks target exploration. In contrast, the composite benchmarks are utilised to assess the algorithm's performances due to their complex nature.

The basic implementation of FA and the proposed CFAEE-SCA algorithms have been validated against five recent cutting-edge metaheuristics tested on the same benchmark function set. The competitor metaheuristics included practical genetic algorithm (RGA) (*Haupt & Haupt, 2004*), gravitational search algorithm (GSA) (*Rashedi & Nezamabadi-pour, 2012*), disruption gravitational search algorithm (D-GSA) (*Sarafrazi, Nezamabadi-pour & Saryazdi, 2011*), clustered gravitational search algorithm (BH-GSA) (*Shams, Rashedi & Hakimi, 2015*), and attractive repulsive gravitational search algorithm (AR-GSA) (*Zandevakili, Rashedi & Mahani, 2019*). The introduced CFAEE-SCA method has been tested in the same way as proposed in *Zandevakili, Rashedi & Mahani*

**Table 1 CEC2013 functions used in the benchmark experiments.**

| No | Functions | Initial range |
|---|---|---|
| Uni-modal Functions | | |
| 1 | Sphere function | $[-100, 100]^D$ |
| 2 | Rotated High Conditioned Elliptic Function | $[-100, 100]^D$ |
| 3 | Rotated Bent Cigar Function | $[-100, 100]^D$ |
| 4 | Rotated Discus Function | $[-100, 100]^D$ |
| 5 | Different Powers Function | $[-100, 100]^D$ |
| Basic multi-modal Functions | | |
| 6 | Rotated Rosenbrock's Function | $[-100, 100]^D$ |
| 7 | Rotated Schaffer's F7 Function | $[-100, 100]^D$ |
| 8 | Rotated Ackley's Function | $[-100, 100]^D$ |
| 9 | Rotated Weierstrass Function | $[-100, 100]^D$ |
| 10 | Rotated Griewank's Function | $[-100, 100]^D$ |
| 11 | Rastrigin's Function | $[-100, 100]^D$ |
| 12 | Rotated Rastrigin's Function | $[-100, 100]^D$ |
| 13 | Non-Continuous Rotated Rastrigin's Function | $[-100, 100]^D$ |
| 14 | Schwefel's Function | $[-100, 100]^D$ |
| 15 | Rotated Schwefel's Function | $[-100, 100]^D$ |
| 16 | Rotated Katsuura Function | $[-100, 100]^D$ |
| 17 | Lunacek Bi_Rastrigin Function | $[-100, 100]^D$ |
| 18 | Rotated Lunacek Bi_Rastrigin Function | $[-100, 100]^D$ |
| 19 | Expanded Griewank's plus Rosenbrock's Function | $[-100, 100]^D$ |
| 20 | Expanded Schaffer's F6 Function | $[-100, 100]^D$ |
| Composite Functions | | |
| 21 | Composite Function 1 ($n = 5$, Rotated) | $[-100, 100]^D$ |
| 22 | Composite Function 2 ($n = 3$, Unrotated) | $[-100, 100]^D$ |
| 23 | Composite Function 3 ($n = 3$, Rotated) | $[-100, 100]^D$ |
| 24 | Composite Function 4 ($n = 3$, Rotated) | $[-100, 100]^D$ |
| 25 | Composite Function 5 ($n = 3$, Rotated) | $[-100, 100]^D$ |
| 26 | Composite Function 6 ($n = 5$, Rotated) | $[-100, 100]^D$ |
| 27 | Composite Function 7 ($n = 5$, Rotated) | $[-100, 100]^D$ |
| 28 | Composite Function 8 ($n = 5$, Rotated) | $[-100, 100]^D$ |

(2019). That publication was utilised to reference the results of other methods included in the comparative analysis. Authors *Zandevakili, Rashedi & Mahani (2019)* proposed a novel version of the GSA by adding the attracting and repulsing parameters to enhance both diversification and intensification phases. It is worth noting that the authors have implemented all algorithms used by *Zandevakili, Rashedi & Mahani (2019)* on their own and tested them independently by using the same experimental setup proposed by *Zandevakili, Rashedi & Mahani (2019)*. The novel CFAEE-SCA has been implemented and verified on all 28 benchmark functions with 30 dimensions ($D = 30$), together with the basic FA implementation.

**Table 2  Results comparison CEC2013 unimodal functions 1–5.**

|  | FA | RGA | GSA | D-GSA | BH-GSA | C-GSA | AR-GSA | CFAEE-SCA |
|---|---|---|---|---|---|---|---|---|
| F1 |  |  |  |  |  |  |  |  |
| Best | 0.00E+00 | 1.845E+02 | 0.00E+00 | 6.71E−01 | 4.57E−13 | 2.28E−13 | 0.00E+00 | 0.00E+00 |
| Median | 0.00E+00 | 2.82E+02 | 0.00E+00 | 9.54E−01 | 3.67E−12 | 2.24E−13 | 0.00E+00 | 0.00E+00 |
| Worst | 2.14E−13 | 3.55E+02 | 2.24E−13 | 1.48E+00 | 5.02E−12 | 4.53E−13 | 0.00E+00 | 0.00E+00 |
| Mean | 6.92E−14 | 2.85E+02 | 7.54E−14 | 9.76E−01 | 3.33E−12 | 2.74E−13 | 0.00E+00 | 0.00E+00 |
| Std | 1.13E−13 | 3.12E+01 | 1.09E−13 | 1.95E−01 | 1.02E−12 | 9.48E−14 | 0.00E+00 | 0.00E+00 |
| F2 |  |  |  |  |  |  |  |  |
| Best | 9.14E+05 | 1.06E+07 | 9.23E+05 | 7.29E+06 | 5.26E+05 | 9.66E+05 | 1.58E+05 | **1.21E+05** |
| Median | 1.75E+06 | 1.61E+07 | 1.72E+06 | 1.14E+07 | 1.95E+06 | 1.77E+06 | 6.08E+05 | **5.75E+05** |
| Worst | 3.59E+06 | 2.52E+07 | 3.36E+06 | 1.84E+07 | 4.92E+06 | **3.10E+06** | 6.55E+06 | 6.51E+06 |
| Mean | 1.26E+06 | 1.72E+07 | 1.84E+06 | 1.18E+07 | 2.03E+06 | 1.85E+06 | 1.39E+06 | **1.18E+06** |
| Std | 5.24E+05 | 3.65E+06 | 5.14E+05 | 2.19E+06 | 7.82E+05 | **4.52E+05** | 1.70E+06 | 1.48E+06 |
| F3 |  |  |  |  |  |  |  |  |
| Best | 2.73E+07 | 3.32E+09 | 2.81E+07 | 1.04E+09 | 4.85E−05 | 2.88E+07 | 7.73E−12 | **7.34E−12** |
| Median | 7.69E+08 | 6.27E+09 | 7.88E+08 | 2.92E+09 | 1.59E+06 | 1.09E+09 | 1.24E−11 | **1.13E−11** |
| Worst | 2.95E+09 | 2.34E+10 | 2.98E+09 | 9.24E+09 | 2.91E+19 | 4.42E+09 | 1.48E−11 | **1.42E−11** |
| Mean | 9.85E+08 | 6.74E+09 | 9.86E+08 | 3.53E+09 | 5.72E+17 | 1.23E+09 | 1.18E−11 | **1.05E−11** |
| Std | 7.54E+08 | 3.01E+09 | 7.16E+08 | 1.74E+09 | 4.09E+18 | 8.44E+08 | 1.83E−12 | **1.74E−12** |
| F4 |  |  |  |  |  |  |  |  |
| Best | 5.75E+04 | 5.18E+04 | 5.74E+04 | 5.82E+04 | 4.96E+04 | 5.62E+04 | 4.61E+04 | **4.39E+04** |
| Median | 6.83E+04 | 7.14E+04 | 6.89E+04 | 6.85E+04 | 6.84E+04 | 6.93E+04 | 6.51E+04 | **6.12E+04** |
| Worst | 7.95E+04 | 1.06E+05 | 7.94E+04 | **7.11E+04** | 9.05E+04 | 8.59E+04 | 7.81E+04 | 7.55E+04 |
| Mean | 6.84E+04 | 7.32E+04 | 6.82E+04 | 6.75E+04 | 6.82E+04 | 7.04E+04 | 6.48E+04 | **6.19E+04** |
| Std | 5.82E+03 | 1.22E+04 | 5.63E+03 | **3.34E+03** | 8.19E+03 | 5.22E+03 | 7.83E+03 | 7.48E+03 |
| F5 |  |  |  |  |  |  |  |  |
| Best | 1.62E−12 | 1.93E+02 | **1.46E−12** | 2.70E+00 | 1.94E−11 | 1.42E−11 | 2.04E−08 | 2.28E−08 |
| Median | 2.64E−12 | 3.05E+02 | **2.41E−12** | 1.51E+01 | 1.02E−10 | 2.12E−11 | 9.93E−08 | 8.85E−08 |
| Worst | 3.98E−12 | 4.63E+02 | **3.74E−12** | 6.04E+01 | 3.33E−10 | 5.74E−11 | 1.85E−07 | 2.89E−07 |
| Mean | 2.65E−12 | 3.04E+02 | **2.38E−12** | 1.90E+01 | 1.23E−10 | 2.34E−11 | 1.05E−07 | 1.53E−07 |
| Std | 5.79E−13 | 6.11E+01 | **5.39E−13** | 1.12E+01 | 7.13E−11 | 7.50E−12 | 3.54E−08 | 3.49E−08 |

**Note:**
The best obtained results for each metric are marked in bold.

In Tables 2–4, the results of the CFAEE-SCA on CEC2013 instances with 30 dimensions and 51 independent runs for uni-modal, multi-modal and composite functions, respectively, have been evaluated against six other swarm intelligence metaheuristics. As mentioned before, the same simulation conditions were utilised as in (*Zandevakili, Rasheedi & Mahani, 2019*), with the same stop criteria of the number of fitness functions evaluations with the maximum number being 1.00E+05. Furthermore, the experiments have been conducted with 50 solutions in the population ($N = 50$).

Convergence graphs of the proposed CFAEE-SCA method for two unimodal, four multimodal and two composite functions that were chosen as examples have been

**Table 3 Results comparison CEC2013 multimodal functions 6–20.**

|  | FA | RGA | GSA | D-GSA | BH-GSA | C-GSA | AR-GSA | CFAEE-SCA |
|---|---|---|---|---|---|---|---|---|
| **F6** | | | | | | | | |
| Best | 2.73E−01 | 7.79E+01 | 2.51E−01 | 5.58E−01 | 2.25E−01 | **1.46E−01** | 3.72E−01 | 2.74E−01 |
| Median | 5.61E+01 | 1.12E+02 | 5.70E+01 | 7.16E+01 | **3.33E+00** | 5.46E+01 | 1.74E+01 | 1.49E+01 |
| Worst | 9.56E+01 | 1.34E+02 | 9.46E+01 | 1.33E+02 | **6.82E+01** | 1.03E+02 | 8.15E+01 | 7.59E+01 |
| Mean | 5.41E+01 | 1.14E+02 | 5.21E+01 | 7.39E+01 | **2.24E+01** | 5.16E+01 | 3.39E+01 | 3.08E+01 |
| Std | 2.71E+01 | **1.19E+01** | 2.53E+01 | 2.48E+01 | 2.71E+01 | 2.48E+01 | 2.68E+01 | 2.34E+01 |
| **F7** | | | | | | | | |
| Best | 2.75E+01 | 4.12E+01 | 2.78E+01 | 3.57E+01 | 4.48E−05 | 3.08E+01 | 4.33E−09 | **4.03E−09** |
| Median | 4.62E+01 | 5.61E+01 | 4.43E+01 | 5.54E+01 | 5.28E−01 | 4.34E+01 | 2.62E−05 | **2.18E−05** |
| Worst | 8.72E+01 | 6.83E+01 | 8.52E+01 | 9.08E+01 | 2.84E+01 | 7.41E+01 | 3.68E−03 | **3.01E−03** |
| Mean | 5.05E+01 | 5.62E+01 | 4.68E+01 | 5.69E+01 | 5.64E+00 | 4.64E+01 | 1.53E−04 | **1.10E−04** |
| Std | 1.58E+01 | 5.64E+00 | 1.22E+01 | 1.23E+01 | 7.65E+00 | 1.12E+01 | 5.24E−04 | **4.92E−04** |
| **F8** | | | | | | | | |
| Best | 2.15E+01 | 2.12E+01 | 2.10E+01 | 2.10E+01 | 2.10E+01 | 2.10E+01 | 2.08E+01 | **1.72E+01** |
| Median | 2.22E+01 | 2.10E+01 | 2.10E+01 | 2.10E+01 | 2.10E+01 | 2.11E+01 | 2.10E+01 | **1.88E+01** |
| Worst | 2.29E+01 | 2.11E+01 | 2.11E+01 | 2.11E+01 | 2.11E+01 | 2.15E+01 | 2.11E+01 | **2.08E+01** |
| Mean | 2.20E+01 | 2.10E+01 | 2.10E+01 | 2.10E+01 | 2.10E+01 | 2.13E+01 | 2.10E+01 | **1.82E+01** |
| Std | 5.63E−02 | 4.69E−02 | 4.81E−02 | 5.32E−02 | 5.64E−02 | 1.61E−01 | 7.15E−02 | **4.49E−02** |
| **F9** | | | | | | | | |
| Best | 2.44E+01 | 1.61E+01 | 2.12E+01 | 2.09E+01 | 3.26E+00 | 2.01E+01 | 2.37E−07 | **2.12E−07** |
| Median | 3.02E+01 | 2.12E+01 | 2.76E+01 | 3.04E+01 | 7.18E+00 | 2.84E+01 | 5.02E+00 | **4.33E+00** |
| Worst | 3.96E+01 | 2.69E+01 | 3.49E+01 | 3.78E+01 | 1.49E+01 | 3.70E+01 | 8.92E+00 | **8.91E+00** |
| Mean | 2.90E+01 | 2.14E+01 | 2.79E+01 | 3.03E+01 | 7.82E+00 | 2.85E+01 | 5.23E+00 | **5.09E+00** |
| Std | 3.73E+00 | 2.34E+00 | 3.55E+00 | 3.93E+00 | 2.46E+00 | 3.62E+00 | 1.97E+00 | **1.83E+00** |
| **F10** | | | | | | | | |
| Best | 0.00E+00 | 3.56E+01 | 0.00E+00 | 1.23E+00 | 5.70E−13 | 3.39E−13 | 0.00E+00 | 0.00E+00 |
| Median | 5.48E−14 | 5.94E+01 | 5.70E−14 | 1.51E+00 | 1.20E−12 | 7.39E−03 | 0.00E+00 | 0.00E+00 |
| Worst | 2.19E−02 | 6.99E+01 | 2.20E−02 | 2.19E+00 | 1.74E−02 | 2.98E−02 | 1.52E−02 | **1.29E−02** |
| Mean | 5.68E−03 | 5.89E+01 | 5.56E−03 | 1.61E+00 | 2.54E−03 | 7.42E−03 | 1.72E−03 | **1.38E−03** |
| Std | 6.59E−03 | 6.74E+00 | 6.41E−03 | 2.72E−01 | 5.03E−03 | 6.05E−03 | 3.85E−03 | **3.52E−03** |
| **F11** | | | | | | | | |
| Best | 1.42E+02 | 1.12E+02 | 1.31E+02 | 1.29E+02 | 8.97E+00 | 1.41E+02 | 7.98E+00 | **7.69E+00** |
| Median | 1.83E+02 | 1.46E+02 | 1.85E+02 | 1.87E+02 | **1.70E+01** | 1.83E+02 | 1.81E+01 | 1.86E+01 |
| Worst | 2.66E+02 | 1.64E+02 | 2.33E+02 | 2.29E+02 | 3.40E+01 | 2.36E+02 | 2.99E+01 | **2.72E+01** |
| Mean | 1.99E+02 | 1.46E+02 | 1.90E+02 | 1.86E+02 | 1.80E+01 | 1.85E+02 | 1.85E+01 | **1.72E+01** |
| Std | 2.55E+01 | 9.18E+00 | 2.38E+01 | 2.20E+01 | 5.19E+00 | 2.14E+01 | 4.50E+00 | **4.23E+00** |
| **F12** | | | | | | | | |
| Best | 1.73E+02 | 1.44E+02 | 1.59E+02 | 1.54E+02 | **7.95E+00** | 1.49E+02 | 1.32E+01 | 1.08E+01 |
| Median | 2.19E+02 | 1.60E+02 | 2.11E+02 | 2.11E+02 | **1.38E+01** | 2.04E+02 | 2.32E+01 | 2.08E+01 |
| Worst | 2.74E+02 | 1.73E+02 | 2.61E+02 | 2.64E+02 | **2.50E+01** | 2.64E+02 | 3.89E+01 | 3.59E+01 |
| Mean | 2.24E+02 | 1.59E+02 | 2.09E+02 | 2.11E+02 | **1.44E+01** | 2.08E+02 | 2.34E+01 | 2.19E+01 |
| Std | 2.89E+01 | 8.68E+00 | 2.74E+01 | 2.41E+01 | **3.75E+00** | 2.38E+01 | 5.43E+00 | 5.28E+00 |

*(Continued)*

|  | FA | RGA | GSA | D-GSA | BH-GSA | C-GSA | AR-GSA | CFAEE-SCA |
|---|---|---|---|---|---|---|---|---|
| **F13** | | | | | | | | |
| Best | 2.52E+02 | 1.32E+02 | 2.77E+02 | 2.48E+02 | 5.15E+00 | 2.42E+02 | 1.17E+01 | **4.83E+00** |
| Median | 3.55E+02 | 1.60E+02 | 3.29E+02 | 3.24E+02 | 2.52E+01 | 3.35E+02 | 4.12E+01 | **2.13E+00** |
| Worst | 4.61E+02 | 1.70E+02 | 4.31E+02 | 4.26E+02 | 6.16E+01 | 4.08E+02 | 8.76E+01 | **6.05E+01** |
| Mean | 3.61E+02 | 1.59E+02 | 3.32E+02 | 3.29E+02 | 2.78E+01 | 3.31E+02 | 4.49E+01 | **2.39E+01** |
| Std | 3.51E+01 | 7.03E+00 | 3.34E+01 | 3.80E+01 | 1.30E+01 | 3.97E+01 | 1.81E+01 | **6.52E+00** |
| **F14** | | | | | | | | |
| Best | 2.24E+03 | 4.38E+03 | 2.21E+03 | 2.48E+03 | 1.07E+03 | 2.19E+03 | 7.82E+02 | **7.15E+02** |
| Median | 3.56E+03 | 5.02E+03 | 3.27E+03 | 3.39E+03 | 1.66E+03 | 3.31E+03 | 1.48E+03 | **1.23E+03** |
| Worst | 4.42E+03 | 5.62E+03 | 4.31E+03 | 4.32E+03 | 2.59E+03 | 4.57E+03 | 2.49E+03 | **2.29E+03** |
| Mean | 3.52E+03 | 5.08E+03 | 3.33E+03 | 3.35E+03 | 1.65E+03 | 3.43E+03 | 1.51E+03 | **1.31E+03** |
| Std | 4.98E+02 | 2.64E+02 | 5.02E+02 | 4.22E+02 | 3.26E+02 | 4.86E+02 | 3.78E+02 | **2.52E+02** |
| **F15** | | | | | | | | |
| Best | 2.52E+03 | 4.57E+03 | 2.41E+03 | 2.15E+03 | 5.14E+02 | 2.31E+03 | 5.32E+02 | **4.99E+02** |
| Median | 3.45E+03 | 5.29E+03 | 3.28E+03 | 3.34E+03 | 1.22E+03 | 3.17E+03 | 1.19E+03 | **1.02E+03** |
| Worst | 4.93E+03 | 5.96E+03 | 4.70E+03 | 5.00E+03 | 2.26E+03 | 4.12E+03 | **1.81E+03** | 2.29E+03 |
| Mean | 3.56E+03 | 5.30E+03 | 3.34E+03 | 3.39E+03 | 1.24E+03 | 3.32E+03 | 1.24E+03 | **1.13E+03** |
| Std | 5.76E+02 | 2.89E+02 | 5.44E+02 | 4.95E+02 | 3.88E+02 | 4.53E+02 | 3.32E+02 | **2.73E+02** |
| **F16** | | | | | | | | |
| Best | 4.35E−04 | 1.94E+00 | **4.09E−04** | 7.00E−01 | 6.06E−04 | 6.05E−04 | 5.52E−04 | 5.34E−04 |
| Median | 2.32E−03 | 2.49E+00 | 2.12E−03 | 1.14E+00 | 3.31E−03 | 2.58E−03 | **2.05E−03** | 2.23E−03 |
| Worst | 9.68E−03 | 3.04E+00 | 9.41E−03 | 1.74E+00 | 1.15E−02 | **9.31E−03** | 1.03E−02 | 1.32E−02 |
| Mean | 2.82E−03 | 2.47E+00 | 2.85E−03 | 1.13E+00 | 3.99E−03 | 3.46E−03 | 2.74E−03 | **2.52E−03** |
| Std | 2.39E−03 | 2.74E−01 | 2.18E−03 | 2.25E−01 | 2.30E−03 | 2.26E−03 | 1.86E−03 | **1.31E−03** |
| **F17** | | | | | | | | |
| Best | 3.74E+01 | 1.93E+02 | 3.75E+01 | 7.44E+01 | 3.72E+01 | 3.61E+01 | 4.09E+01 | **3.39E+01** |
| Median | 4.35E+01 | 2.10E+02 | 4.45E+01 | 1.03E+02 | 4.59E+01 | **4.32E+01** | 5.03E+01 | 4.56E+01 |
| Worst | 6.68E+01 | 2.34E+02 | 6.72E+01 | 1.26E+02 | **5.65E+01** | 5.72E+01 | 6.52E+01 | 7.69E+01 |
| Mean | 4.73E+01 | 2.12E+02 | 4.49E+01 | 1.03E+02 | 4.64E+01 | **4.42E+01** | 5.03E+01 | 4.52E+01 |
| Std | 5.45E+00 | 9.39E+00 | 5.06E+00 | 1.09E+01 | 4.11E+00 | 4.39E+00 | 5.29E+00 | **3.99E+00** |
| **F18** | | | | | | | | |
| Best | 3.85E+01 | 1.84E+02 | 3.65E+01 | 1.34E+02 | 3.95E+01 | 3.74E+01 | 4.15E+01 | **3.53E+01** |
| Median | 4.74E+01 | 2.11E+02 | 4.54E+01 | 1.72E+02 | 4.72E+01 | **4.44E+01** | 5.53E+01 | 4.48E+01 |
| Worst | 5.93E+01 | 2.30E+02 | 5.36E+01 | 1.98E+02 | 5.92E+01 | 5.88E+01 | 7.13E+01 | **5.09E+01** |
| Mean | 4.86E+01 | 2.11E+02 | 4.54E+01 | 1.75E+02 | 4.72E+01 | 4.58E+01 | 5.59E+01 | **4.42E+01** |
| Std | 3.92E+00 | 8.89E+00 | 3.76E+00 | 1.43E+01 | 4.04E+00 | 4.24E+00 | 7.11E+00 | **3.62E+00** |
| **F19** | | | | | | | | |
| Best | 1.75E+00 | 2.17E+01 | 1.79E+00 | 4.33E+00 | 2.77E+00 | **1.72E+00** | 2.56E+00 | 2.29E+00 |
| Median | 2.69E+00 | 2.54E+01 | **2.75E+00** | 6.48E+00 | 4.59E+00 | 3.04E+00 | 3.55E+00 | 3.63E+00 |
| Worst | 4.08E+00 | 2.92E+01 | **4.38E+00** | 1.55E+01 | 6.25E+00 | 4.44E+00 | 6.86E+00 | 6.89E+00 |
| Mean | 2.82E+00 | 2.53E+01 | **2.96E+00** | 7.26E+00 | 4.69E+00 | 3.02E+00 | 3.85E+00 | 3.89E+00 |
| Std | 6.82E−01 | 1.59E+00 | 6.78E−01 | 2.76E+00 | 9.54E−01 | **6.27E−01** | 8.87E−01 | 8.49E−01 |

|  | FA | RGA | GSA | D-GSA | BH-GSA | C-GSA | AR-GSA | CFAEE-SCA |
|---|---|---|---|---|---|---|---|---|
| **F20** | | | | | | | | |
| Best | 1.66E+01 | 1.50E+01 | 1.42E+01 | **1.41E+01** | 1.50E+01 | 1.50E+01 | 1.48E+01 | 1.45E+01 |
| Median | 1.67E+01 | 1.50E+01 | 1.50E+01 | 1.50E+01 | 1.50E+01 | 1.50E+01 | 1.50E+01 | 1.50E+01 |
| Worst | 1.69E+01 | 1.50E+01 | 1.50E+01 | 1.50E+01 | 1.50E+01 | 1.50E+01 | 1.50E+01 | 1.50E+01 |
| Mean | 1.68E+01 | 1.50E+01 | 1.50E+01 | 1.50E+01 | 1.50E+01 | 1.50E+01 | 1.50E+01 | 1.50E+01 |
| Std | 1.46E−01 | 9.94E−06 | 1.34E−01 | 1.83E−01 | 6.29E−08 | 3.11E−06 | 1.99E−02 | **6.11E−08** |

**Note:**
The best obtained results for each metric are marked in bold.

**Table 4 Results comparison CEC2013 composite functions 21–28.**

|  | FA | RGA | GSA | D-GSA | BH-GSA | C-GSA | AR-GSA | CFAEE−SCA |
|---|---|---|---|---|---|---|---|---|
| **F21** | | | | | | | | |
| Best | 1.33E+02 | 4.61E+02 | 1.00E+02 | 1.28E+02 | 2.01E+02 | **1.00E+02** | 2.00E+02 | 1.93E+02 |
| Median | 3.68E+02 | 5.64E+02 | 3.00E+02 | 3.16E+02 | 3.00E+02 | 3.00E+02 | 3.00E+02 | **2.79E+02** |
| Worst | 4.79E+02 | 6.05E+02 | 4.44E+02 | 4.44E+02 | 4.44E+02 | 4.44E+02 | 4.44E+02 | **4.24E+02** |
| Mean | 3.39E+02 | 5.39E+02 | 3.18E+02 | 3.39E+02 | 3.35E+02 | 3.34E+02 | 3.25E+02 | **3.09E+02** |
| Std | 7.45E+01 | 4.31E+01 | 7.27E+01 | 7.16E+01 | 9.11E+01 | 7.99E+01 | 9.24E+01 | **4.24E+01** |
| **F22** | | | | | | | | |
| Best | 3.99+03 | 4.32E+03 | 3.79E+03 | 4.01E+03 | 3.30E+02 | 3.88E+03 | 3.10E+02 | **3.04E+02** |
| Median | 5.49E+03 | 4.98E+03 | 5.20E+03 | 5.41E+03 | 1.09E+03 | 5.54E+03 | 1.12E+03 | **1.04E+03** |
| Worst | 7.33E+03 | 5.76E+03 | 7.12E+03 | 7.06E+03 | 2.23E+03 | 7.51E+03 | 2.25E+03 | **2.14E+03** |
| Mean | 5.66E+03 | 5.07E+03 | 5.36E+03 | 5.55E+03 | 1.21E+03 | 5.53E+03 | 1.11E+03 | **1.02E+03** |
| Std | 8.96E+02 | 3.42E+02 | 8.60E+02 | 7.92E+02 | 4.11E+02 | 8.05E+02 | 3.85E+02 | **3.19E+02** |
| **F23** | | | | | | | | |
| Best | 4.45E+03 | 4.39E+03 | 4.24E+03 | 4.87E+03 | **6.02E+02** | 3.85E+03 | 1.02E+03 | 1.24E+03 |
| Median | 5.73E+03 | 5.42E+03 | 5.51E+03 | 5.57E+03 | 1.97E+03 | 5.50E+03 | **1.85E+03** | 1.89E+03 |
| Worst | 6.95E+03 | 6.23E+03 | 6.68E+03 | 6.42E+03 | 4.26E+03 | 6.11E+03 | **3.77E+03** | 3.84E+03 |
| Mean | 5.83E+03 | 5.42E+03 | 5.56E+03 | 5.60E+03 | 2.12E+03 | 5.46E+03 | **1.96E+03** | 2.12E+03 |
| Std | 4.62E+02 | 4.04E+02 | 4.38E+02 | 3.24E+02 | 7.62E+02 | 4.33E+02 | 6.02E+02 | **3.09E+02** |
| **F24** | | | | | | | | |
| Best | 2.55E+02 | 2.32E+02 | 2.20E+02 | 2.17E+02 | 2.02E+02 | 2.30E+02 | 2.00E+02 | **1.98E+02** |
| Median | 2.75E+02 | 2.38E+02 | 2.59E+02 | 2.60E+02 | 2.01E+02 | 2.57E+02 | 2.00E+02 | **1.99E+02** |
| Worst | 3.99E+02 | 2.79E+02 | 3.92E+02 | 3.83E+02 | 2.11E+02 | 3.88E+02 | **2.00E+02** | 2.05E+02 |
| Mean | 2.86E+02 | 2.41E+02 | 2.80E+02 | 2.72E+02 | 2.02E+02 | 2.69E+02 | 2.00E+02 | 2.00E+00 |
| Std | 4.65E+01 | 1.13E+01 | 4.50E+01 | 3.77E+01 | 1.19E−01 | 3.64E+01 | 2.46E−02 | **2.15E−02** |
| **F25** | | | | | | | | |
| Best | 2.30E+02 | 2.41E+02 | 2.00E+02 | 2.10E+02 | 2.00E+02 | 2.00E+02 | 2.00E+02 | **1.88E+02** |
| Median | 3.62E+02 | 2.84E+02 | 3.42E+02 | 3.49E+02 | 2.00E+02 | 3.40E+02 | 2.00E+02 | **1.94E+02** |
| Worst | 4.23E+02 | 3.05E+02 | 3.88E+02 | 3.87E+02 | 2.72E+02 | 3.84E+02 | 2.00E+02 | **1.98E+02** |
| Mean | 3.91E+02 | 2.73E+02 | 3.34E+02 | 3.40E+02 | 2.13E+02 | 3.34E+02 | 2.00E+02 | **1.95E+02** |
| Std | 4.62E+01 | 2.52E+01 | 4.08E+01 | 3.70E+01 | 2.54E+01 | 4.18E+01 | 1.84E−05 | **1.79E−05** |

(Continued)

|  | FA | RGA | GSA | D-GSA | BH-GSA | C-GSA | AR-GSA | CFAEE–SCA |
|---|---|---|---|---|---|---|---|---|
| **F26** | | | | | | | | |
| Best | 2.62E+02 | 2.21E+02 | 2.35E+02 | 2.00E+02 | **1.11E+02** | 2.00E+02 | 2.28E+02 | 2.29E+02 |
| Median | 3.73E+02 | 3.41E+02 | 3.42E+02 | 3.51E+02 | 3.00E+02 | 3.47E+02 | **2.97E+02** | 3.02E+02 |
| Worst | 3.98E+02 | 3.65E+02 | 3.77E+02 | 3.71E+02 | 3.26E+02 | 3.73E+02 | **3.20E+02** | 3.23E+02 |
| Mean | 3.48E+02 | 3.16E+02 | 3.30E+02 | 3.34E+02 | **2.86E+02** | 3.25E+02 | 2.93E+02 | 2.96E+02 |
| Std | 3.94E+01 | 6.05E+01 | 3.76E+01 | 4.37E+01 | 4.29E+01 | 4.78E+01 | 1.72E+01 | **1.66E+01** |
| **F27** | | | | | | | | |
| Best | 5.75E+02 | 6.18E+02 | 5.85E+02 | 6.12E+02 | 3.01E+02 | 6.26E+02 | **3.00E+02** | 3.18E+02 |
| Median | 7.58E+02 | 7.95E+02 | 7.64E+02 | 8.42E+02 | 3.01E+02 | 7.69E+02 | **3.00E+02** | 3.22E+02 |
| Worst | 9.89E+02 | 1.03E+03 | 9.88E+02 | 1.04E+03 | 3.05E+02 | 1.02E+03 | **3.03E+02** | 3.29E+02 |
| Mean | 7.83E+02 | 7.76E+02 | 7.88E+02 | 8.40E+02 | 3.02E+02 | 7.86E+02 | **3.00E+02** | 3.23E+02 |
| Std | 1.06E+02 | 1.35E+02 | 1.10E+02 | 1.12E+02 | 1.15E+00 | 9.31E+01 | **4.11E−01** | 4.29E−01 |
| **F28** | | | | | | | | |
| Best | 2.42E+03 | 5.10E+02 | 2.47E+03 | 2.84E+03 | **1.00E+02** | 2.35E+03 | 3.00E+02 | 3.25E+02 |
| Median | 3.35E+03 | 8.14E+02 | 3.12E+03 | 3.23E+03 | 3.01E+02 | 3.18E+03 | 3.00E+02 | 3.27E+02 |
| Worst | 3.67E+03 | 1.76E+03 | 3.69E+03 | 3.93E+03 | 1.37E+03 | 3.94E+03 | **3.00E+02** | 3.33E+02 |
| Mean | 3.10E+03 | 8.93E+02 | 3.16E+03 | 3.24E+03 | 3.52E+02 | 3.25E+03 | **3.00E+02** | 3.26E+02 |
| Std | 2.66E+02 | 3.54E+02 | 2.73E+02 | 2.38E+02 | 2.54E+02 | 2.92E+02 | **8.81E−09** | 8.95E−09 |

**Note:**
The best obtained results for each metrics are marked in bold.

presented in Fig. 2. The proposed CFAEE-SCA has been compared to the basic FA, and cutting-edge metaheuristics such as AR-GSA, GSA and RGA. From the presented convergence graphs, it can be seen that the proposed method in most cases converges faster than the other metaheuristics included in the experiments. Additionally, the proposed method is significantly superior to the basic FA metaheuristics, that in most cases stagnates while the CFAEE-SCA accelerates the convergence speed.

In order to provide more objective way for determining the performances and efficiency of the proposed method against other competitors, statistical tests must be conducted. Therefore, the Friedman test that was introduced by *Friedman (1937, 1940)*, together with the ranked two-way analysis of variances of the suggested approach and other implemented algorithms were conducted.

The results obtained by the eight implemented approaches on the set of 28 challenging function instances from the CEC2013 benchmark suite, including the Friedman and the aligned Friedman test, are given in the Tables 5 and 6, respectively.

According to the findings presented in Table 6, the proposed CFAEE-SCA outscored all other algorithms, together with the original FA which achieved the average rank of 133.463. Suggested CFAEE-SCA achieved an average ranking of 56.838.

Additionally, the research by *Sheskin (2020)* suggested the possible enhancement in terms of performance by comparing with the $\chi^2$ value. Therefore, the Iman and Davenport's test introduced by *Iman & Davenport (1980)* has been applied as well. The findings of this test are presented in Table 7.

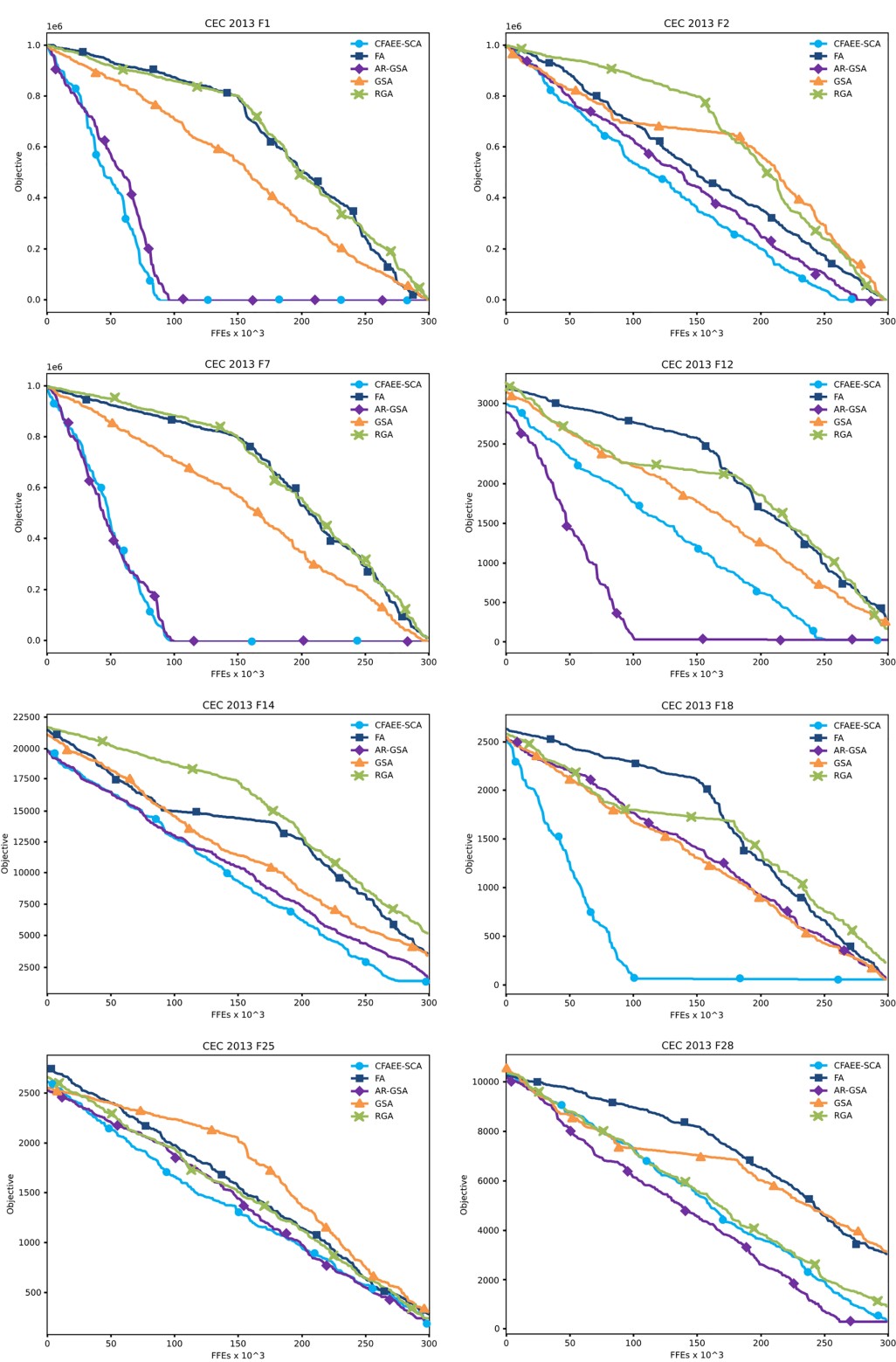

**Figure 2 Converging velocity graphs of the eight CEC 2013 benchmark functions as direct comparison between the proposed CFAEE-SCA method and other relevant algorithms.**

**Table 5 Friedman test ranks for the observed methods over 28 CEC2013 functions.**

| Functions | FA | RGA | GSA | D-GSA | BH-GSA | C-GSA | AR-GSA | CFAEE-SCA |
|---|---|---|---|---|---|---|---|---|
| F1 | 3 | 8 | 4 | 7 | 6 | 5 | 1.5 | 1.5 |
| F2 | 1 | 8 | 5 | 7 | 6 | 4 | 3 | 2 |
| F3 | 4 | 7 | 3 | 6 | 8 | 5 | 2 | 1 |
| F4 | 6 | 8 | 4.5 | 3 | 4.5 | 7 | 2 | 1 |
| F5 | 1 | 8 | 2 | 7 | 4 | 3 | 5 | 6 |
| F6 | 4 | 8 | 6 | 7 | 1 | 5 | 3 | 2 |
| F7 | 6 | 7 | 5 | 8 | 3 | 4 | 2 | 1 |
| F8 | 2 | 5.5 | 5.5 | 5.5 | 5.5 | 8 | 3 | 1 |
| F9 | 7 | 4 | 5 | 8 | 3 | 6 | 2 | 1 |
| F10 | 5 | 8 | 4 | 7 | 3 | 6 | 2 | 1 |
| F11 | 8 | 4 | 7 | 5.5 | 2 | 5.5 | 3 | 1 |
| F12 | 5 | 4 | 7 | 8 | 1 | 6 | 3 | 2 |
| F13 | 8 | 4 | 7 | 5 | 2 | 6 | 3 | 1 |
| F14 | 4 | 8 | 5 | 6 | 3 | 7 | 2 | 1 |
| F15 | 4 | 8 | 6 | 7 | 3 | 5 | 2 | 1 |
| F16 | 4 | 8 | 3 | 7 | 6 | 5 | 2 | 1 |
| F17 | 5 | 8 | 2 | 7 | 4 | 1 | 6 | 3 |
| F18 | 5 | 8 | 2 | 7 | 4 | 3 | 6 | 1 |
| F19 | 2 | 8 | 1 | 7 | 6 | 3 | 4 | 5 |
| F20 | 8 | 4 | 4 | 4 | 4 | 4 | 4 | 4 |
| F21 | 1 | 8 | 3 | 7 | 6 | 5 | 4 | 2 |
| F22 | 8 | 4 | 5 | 7 | 3 | 6 | 2 | 1 |
| F23 | 8 | 4 | 6 | 7 | 2.5 | 5 | 1 | 2.5 |
| F24 | 8 | 4 | 7 | 6 | 3 | 5 | 2 | 1 |
| F25 | 8 | 4 | 5.5 | 7 | 3 | 5.5 | 2 | 1 |
| F26 | 4 | 5 | 7 | 8 | 1 | 6 | 2 | 3 |
| F27 | 7 | 4 | 5.5 | 8 | 2 | 5.5 | 1 | 3 |
| F28 | 5 | 4 | 6 | 8 | 3 | 7 | 1 | 2 |
| Average Ranking | 5.036 | 6.161 | 4.750 | 6.679 | 3.661 | 5.125 | 2.696 | 1.893 |
| Rank | 5 | 7 | 4 | 8 | 3 | 6 | 2 | 1 |

The obtained findings show a value of 2.230E+01 that indicates significantly better results than the $F$-distribution critical value ($F(9,9 \times 10) = 2.058E+00$). Additionally, the null hypothesis $H_0$ has been rejected by Iman and Davenport's test. The Friedman statistics score of ($\chi_r^2 = 1.407E+01$) results in better performance than the $F$-distribution critical value at the level of significance of $\alpha = 0.05$.

The final observation that can be drawn here is that the null hypothesis ($H_0$) can be rejected and that the proposed CFAEE-SCA is obviously the best algorithm in the conducted tests.

As both executed statistical tests rejected the null hypothesis, the next type of test, namely the Holm's step-down procedure has been performed. This procedure is a non-parametric post-hoc method. The results of this procedure have been presented in Table 8.

**Table 6 Aligned Friedman test ranks for the observed methods over 28 CEC2013 functions.**

| Functions | FA | RGA | GSA | D-GSA | BH-GSA | C-GSA | AR-GSA | CFAEE-SCA |
|---|---|---|---|---|---|---|---|---|
| F1 | 64 | 192 | 65 | 68 | 67 | 66 | 62.5 | 62.5 |
| F2 | 8 | 223 | 12 | 222 | 13 | 11 | 10 | 9 |
| F3 | 4 | 7 | 3 | 6 | 224 | 5 | 1.5 | 1.5 |
| F4 | 216 | 221 | 195.5 | 32 | 195.5 | 219 | 15 | 14 |
| F5 | 51 | 194 | 52 | 85 | 54 | 53 | 55 | 56 |
| F6 | 109 | 167 | 115 | 153 | 73 | 112 | 87 | 84 |
| F7 | 148 | 155 | 147 | 157 | 79 | 146 | 71 | 70 |
| F8 | 123 | 132.5 | 132.5 | 132.5 | 132.5 | 136 | 130 | 113 |
| F9 | 144 | 139 | 142 | 145 | 95 | 143 | 93 | 92 |
| F10 | 103 | 164 | 102 | 105 | 101 | 104 | 100 | 99 |
| F11 | 175 | 156 | 170 | 168.5 | 44 | 168.5 | 45 | 43 |
| F12 | 171 | 159 | 173 | 174 | 40 | 172 | 42 | 41 |
| F13 | 190 | 50 | 182 | 179 | 38 | 180 | 39 | 37 |
| F14 | 187 | 218 | 197 | 199 | 30 | 201 | 29 | 27 |
| F15 | 193 | 220 | 200 | 202 | 24 | 198 | 23 | 22 |
| F16 | 127 | 138 | 126 | 137 | 129 | 128 | 125 | 124 |
| F17 | 82 | 183 | 75 | 158 | 77 | 74 | 83 | 76 |
| F18 | 69 | 181 | 59 | 177 | 61 | 60 | 78 | 57 |
| F19 | 107 | 150 | 106 | 135 | 114 | 108 | 110 | 111 |
| F20 | 140 | 119 | 119 | 119 | 119 | 119 | 119 | 119 |
| F21 | 49 | 189 | 72 | 97 | 91 | 89 | 81 | 58 |
| F22 | 217 | 206 | 212 | 214 | 18 | 213 | 17 | 16 |
| F23 | 215 | 203 | 207 | 208 | 20.5 | 204 | 19 | 20.5 |
| F24 | 176 | 152 | 166 | 163 | 88 | 162 | 86 | 36 |
| F25 | 178 | 94 | 160.5 | 165 | 48 | 160.5 | 47 | 46 |
| F26 | 98 | 141 | 151 | 154 | 80 | 149 | 90 | |
| F27 | 188 | 184 | 185.5 | 191 | 34 | 185.5 | 33 | 35 |
| F28 | 205 | 31 | 209 | 211 | 28 | 210 | 25 | 26 |
| Average Ranking | 133.464 | 156.018 | 133.429 | 148.464 | 75.625 | 134.875 | 61.286 | 56.839 |
| Rank | 5 | 8 | 4 | 7 | 3 | 6 | 2 | 1 |

**Table 7 Friedman and Iman-Davenport statistical test results summary ($\alpha = 0.05$).**

| Friedman value | $\chi^2$ critical value | $p$-value | Iman-Davenport value | $F$-critical value |
|---|---|---|---|---|
| 8.866E+01 | 1.407E+01 | 1.110E−16 | 2.230E+01 | 2.058E+00 |

The $p$ value is the main sorting reference for all approaches included in the experiment, and they are compared against the $\alpha/(k - i)$. The $k$ represents the degree of freedom, and the $i$ denotes the number of the method.

This paper used the $\alpha$ parameter at the levels of 0.05 and 0.1. It is worth mentioning that the values of $p$ parameter are given in scientific notation.

**Table 8 Results of the Holm's step-down procedure.**

| Comparison | $p'$ values | Ranking | alpha = 0.05 | alpha = 0.1 | H1 | H2 |
|---|---|---|---|---|---|---|
| CFAEE-SCA *vs* D-GSA | 1.33227E−13 | 0 | 0.007142857 | 0.014285714 | TRUE | TRUE |
| CFAEE-SCA *vs* RGA | 3.53276E−11 | 1 | 0.008333333 | 0.016666667 | TRUE | TRUE |
| CFAEE-SCA *vs* C-GSA | 3.96302E−07 | 2 | 0.01 | 0.02 | TRUE | TRUE |
| CFAEE-SCA *vs* FA | 7.90191E−07 | 3 | 0.0125 | 0.025 | TRUE | TRUE |
| CFAEE-SCA *vs* GSA | 6.37484E−06 | 4 | 0.016666667 | 0.033333333 | TRUE | TRUE |
| CFAEE-SCA *vs* BH-GSA | 0.003462325 | 5 | 0.025 | 0.05 | TRUE | TRUE |
| CFAEE-SCA *vs* AR-GSA | 0.109821937 | 6 | 0.05 | 0.1 | FALSE | FALSE |

The summary of the conducted Holm's procedure presented in the Table 8 indicates that the significant enhancement has been achieved by the proposed method in case of both levels of significance.

# THE XGBOOST CLASSIFIER TUNING WITH CFAEE-SCA

In this section, the basic information relevant to the framework for optimising the XGBoost model by using the proposed CFAEE-SCA algorithm are shown. Later on, this section presents the results of the proposed approach on two sets of network intrusion detection experiments. First experiment was conducted by utilising the NSL-KDD benchmark dataset, while the second experiment used more recent, UNSW-NB15 network intrusion dataset.

## The CFAEE-SCA-XGBoost overview

The XGBoost is an extensible and configurable improved gradient Boosting decision tree optimiser with fast computation and good performance. It constructs Boosted regression and classification trees, which operate in parallel. It efficiently optimises the value of the objective function. According to *Chen & Guestrin (2016)*, it works by scoring the frequency and by measuring the coverage of the impact of a selected feature on the output of a function.

XGBoost utilises additive training optimisation, where each new iteration is dependant on the result of the previous one. This is evident in the $i$-th iteration's objective function calculation method:

$$g_j = \partial_{\hat{y}_k^{i-1}} l\left(y_j, \hat{y}_k^{i-1}\right) \tag{23}$$

$$h_j = \partial_{\hat{y}_k^{i-1}}^2 l\left(y_j, \hat{y}_k^{i-1}\right) \tag{24}$$

$$w_j^* = -\frac{\sum g_t}{\sum h_t + \lambda} \tag{25}$$

$$R(f_i) = \gamma T_i + \frac{\lambda}{2}\sum_{j=1}^{T} w_j^2 \tag{26}$$

$$F_o{}^i = \sum_{k=1}^{n} l\left(y_k, \hat{y}_k^{i-1} + f_i(x_k)\right) + R(f_i) + C \tag{27}$$

In Eqs. (23)–(27), $g$ and $h$ are the 1st and 2nd derivatives, $w$ are the weights, $R$ is the model's regularisation term, $\gamma$ and $\lambda$ are parameters for configuring the tree structure

**Table 9 XGBoost parameters optimised by CFAEE-SCA.**

| Parameter | Default | Range | Details |
|---|---|---|---|
| eta | 0.3 | [0, 1] | Learning rate |
| max_depth | 6 | [0, +∞] | Maximum depth of the tree |
| min_child_weight | 1 | [0, +∞] | Minimum leaf weight |
| gamma | 0 | [0, +∞] | Related to loss function |
| sub-sample | 1 | (0, 1] | Controls sampling to prevent over-fitting |
| colsample_bytree | 1 | (0, 1] | Controls feature sampling proportions |

(larger values give simpler trees). $F_o^i$ is the $i$-th iteration's object function, $l$ is the loss term in that iteration, and $C$ is a constant term. Finally, the score of the loss function, which is used to evaluate the complexity of the tree structure:

$$F_o^* = -\frac{1}{2}\sum_{j=1}^{T}\frac{(\sum g)^2}{\sum h + \lambda} + \gamma T \tag{28}$$

The proposed CFAEE-SCA-XGBoost model's parameters are optimised using the CFAEE-SCA algorithm. The six optimised parameters shown in Table 9. The parameters have been chosen based on the several previous published research including *Jiang et al. (2020)*, as they have the most influence on the performances of the model. The same parameters have been optimised for both conducted experiments.

Therefore, the proposed CFAEE-SCA solution is encoding as a vector with six components, where each vector's parameter represents one XGBoost hyper-parameter from Table 9 which is subject to optimisation process. Some of the components are continuous (eta, gamma,sub-sample,colsample_bytree) and some are integer (max_depth and min_child_weight) and this represents a typical mixed variables NP-hard challenge. During the search process, due to the search expressions of the CFAEE-SCA optimiser, integer variables are transformed to continuous, and they are eventually transformed back to integers by using simple sigmoid transfer function.

The fitness of each solution is calculated by constructing the XGBoost model based on the solution and validating its performance on the training set, while for the global best solution (the one that establishes the best fitness on the training set), the constructed XGBoost model is validated against the testing set and these metrics are reported in the results' tables. Pipeline of the CFAEE-SCA-XGBoost framework is presented in Fig. 3.

## Experiments with NSL-KDD dataset

The proposed model was trained and tested using the NSL-KDD dataset, which was analysed for the first time in *Tavallaee et al. (2009)*. The NSL-KDD dataset can be retrieved from the following URL: https://unb.ca/cic/datasets/nsl.html. This dataset is prepared and used for intrusion Detection system evaluation. Dataset features are described in *Protić (2018)*. A summary describing the main features of the dataset is shown in Table 10. The proposed model was tested with the swarm size of 100 agents throughout 800 iterations, with 8,000 fitness function evaluations (FFE). This setup was proposed by *Jiang et al. (2020)*.

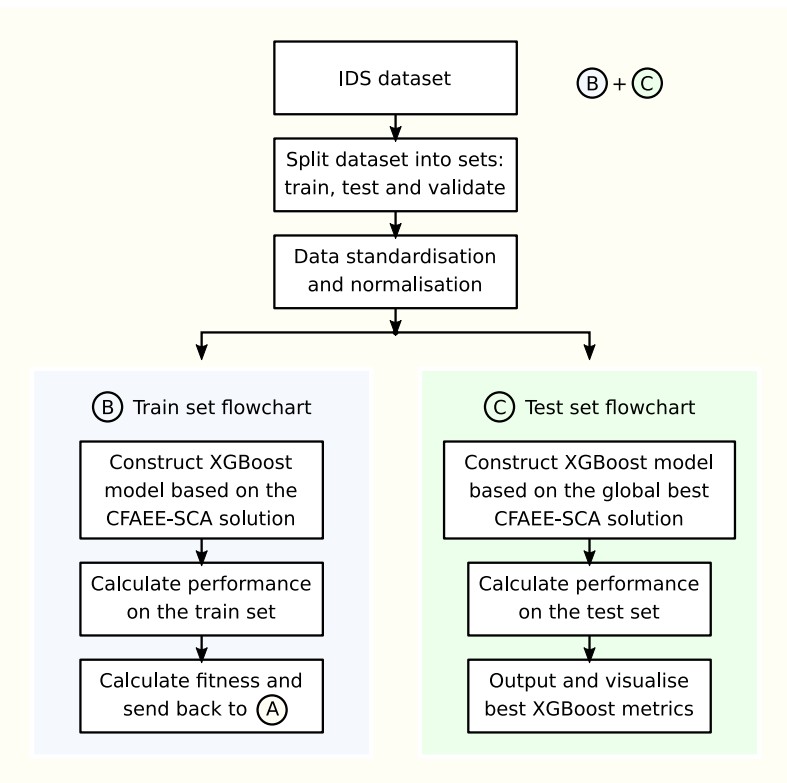

**Figure 3** **Pipeline of CFAEE-SCA-XGBoost framework.**

**Table 10** NSL-KDD dataset summary.

| Property | Description |
| --- | --- |
| Number of records | 126,620 |
| Number of features | 41 |
| Number of classes | 2 (normal uses and attacks) |
| Groups of attacks | 4 (Probe, DoS, U2R and R2L) |
| Types of attacks | 38 in total (21 in training set) |
| Number of sets | 2 (a training and a testing set) |

There are five event classes which represent normal use, denial of service (DoS) attack, probe attack, user to root attack (U2R), and remote to local user (R2L). As very well documented by *Protić (2018)*, the dataset has predefined training and testing sets, whose structure is shown in Table 11, while visual representation is provided in Fig. 4.

The proposed model was tested, following instructions set up by *Jiang et al. (2020)*, with the substituting of their optimisation algorithm with the proposed CFAEE-SCA algorithm, for this experiment.

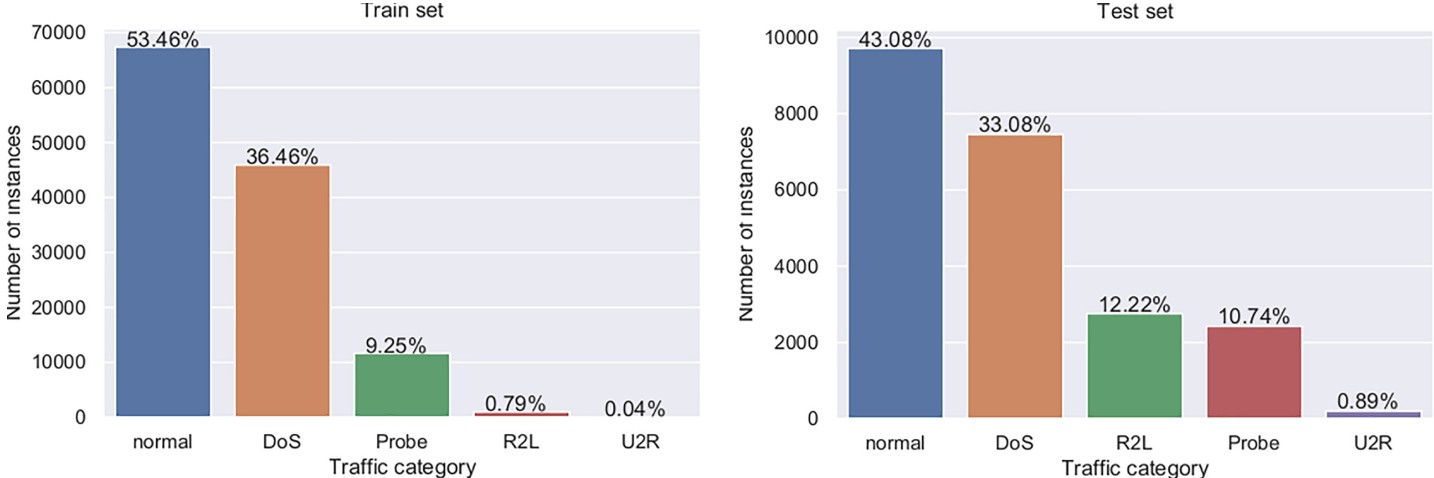

**Figure 4 Visual representation of training and testing NSL-KDD datasets.**

**Table 11 NSL-KDD dataset structure.**

| Event type | Training set | | Testing set | |
|---|---|---|---|---|
| Normal use | 67,343 | 53.46% | 9,711 | 43.08% |
| DoS | 45,927 | 36.46% | 7,456 | 33.07% |
| Probe | 11,656 | 9.25% | 2,421 | 10.74% |
| U2R | 52 | 0.04% | 200 | 0.89% |
| R2L | 995 | 0.79% | 2,756 | 12.22% |
| Total | 125,973 | | 22,544 | |

Because of different types of data in the dataset, data-points are standardised into a continuous range:

$$d'_{ij} = \frac{d_{ij} - \frac{1}{M}\sum_{i=1}^{M} d_{ij}}{\frac{1}{M}\sum_{i=1}^{M}\left| d_{ij} - \frac{1}{M}\sum_{i=1}^{M} d_{ij}\right|} \tag{29}$$

In Eq. (29), $M$ represents the total number of records in the dataset, $d$ is an individual data-point for the $i$-th feature of the $j$-th record, and $d'$ is the corresponding data-point's standardised value. After standardising all data-points, they are normalised:

$$d''_{ij} = \frac{d'_{ij} - d_{min}}{d_{max} - d_{min}} \tag{30}$$

In Eq. (30), $d''$ is the normalised value of the corresponding $d'$ data-point. $d_{min}$ and $d_{max}$ are the minimum and maximum values of the $j$-th feature.

**Table 12 The dataset testing set optimal parameters confusion matrix.**

|  |  | Normal | Probe | Dos | U2R | R2L | Average/total |
|---|---|---|---|---|---|---|---|
| XGBoost | Precision | 0.63 | 0.75 | **0.96** | 0.75 | 0.67 | 0.76 |
|  | Recall | 0.97 | 0.71 | 0.67 | 0.03 | 0.00 | 0.72 |
|  | F-Score | 0.76 | 0.73 | 0.79 | 0.06 | 0.00 | 0.67 |
|  | Support | 9,711 | 2,421 | 7,458 | 200 | 2,754 | 22,544 |
| PSO-XGBoost | Precision | 0.66 | **0.81** | 0.94 | **1.00** | **0.95** | 0.81 |
|  | Recall | 0.96 | 0.52 | 0.84 | 0.01 | 0.05 | 0.74 |
|  | F-Score | 0.76 | 0.64 | 0.87 | 0.01 | 0.09 | 0.70 |
|  | Support | 9,771 | 2,421 | 7,458 | 200 | 2,754 | 22,544 |
| FA-XGBoost | Precision | 0.67 | 0.79 | 0.93 | 0.92 | 0.85 | 0.79 |
|  | Recall | 0.97 | 0.63 | 0.87 | 0.15 | 0.62 | 0.76 |
|  | F-Score | 0.77 | 0.68 | 0.88 | 0.19 | 0.64 | 0.72 |
|  | Support | 9,771 | 2,421 | 7,458 | 200 | 2,754 | 22,544 |
| CFAEE-SCA-XGBoost | Precision | **1.00** | 0.79 | 0.91 | 0.89 | 0.86 | **0.93** |
|  | Recall | **1.00** | **0.92** | 0.91 | 0.21 | 0.79 | **0.93** |
|  | F-Score | **1.00** | **0.85** | 0.91 | 0.34 | 0.82 | **0.93** |
|  | Support | 9,771 | 2,421 | 7,458 | 200 | 2,754 | 22,544 |

**Note:**
The best achieved performance metric in all comparative analysis results tables are marked in bold.

The proposed model is evaluated using precision, recall, f-score, and the P-R curve. The P-R curve is used instead of the ROC curve due to its better ability to capture the binary event situation measurement impact, as explained by *Sofaer, Hoeting & Jarnevich (2019)*. Specifically, these events happen in this dataset due to a limited number of U2R attack cases related to other events. P-R curve-based values, including the average precision (AP), mean average precision (mAP) and macro-averaging calculations, further help evaluate the model's performance.

Experimental results of the proposed model are presented and compared to results of the solution with the pure XGBoost approach, the original FA-XGBoost and the PSO-XGBoost. The experimental setup is the same as the setup proposed in *Jiang et al. (2020)*, that was used to reference the PSO-XGBoost results. It is important to state that the authors have implemented the PSO-XGBoost and tested it independently, by using the same conditions as in *Jiang et al. (2020)*. Results for the FA and CFAEE-SCA supported versions of the XGBoost framework are shown in Table 12, together with the PSO-XGBoost and basic XGBoost results. The best results are marked in bold. As the presented results show, the proposed CFAEE-SCA-XGBoost approach clearly outperforms both other metaheuristics approaches for the observed classes. Additionally, it can be seen that the CFAEE-SCA-XGBoost significantly outperforms the basic XGBoost method. The basic FA-XGBoost obtained similar level of performances as PSO-XGBoost.

Table 13 shows AP values from the P-R curves of the CFAEE-SCA-XGBoost model compared to the values of XGBoost, FA-XGBoost and PSO-XGBoost models for all event types and classes. The proposed CFAEE-SCA-XGBoost approach performed better than other compared approaches for all types and classes. It is important to note that the

**Table 13 Comparison of AP values for each class.**

|  | XGBoost | PSO-XGBoost | FA-XGBoost | CFAEE-SCA-XGBoost |
|---|---|---|---|---|
| Normal | 0.89 | 0.89 | 0.90 | **1.0** |
| Probe | 0.75 | 0.79 | 0.78 | **0.93** |
| Dos | 0.88 | 0.94 | 0.93 | **0.98** |
| U2R | 0.11 | 0.15 | 0.24 | **0.33** |
| R2L | 0.42 | 0.48 | 0.55 | **0.94** |

Note:
The best achieved performance metric in all comparative analysis results tables are marked in bold.

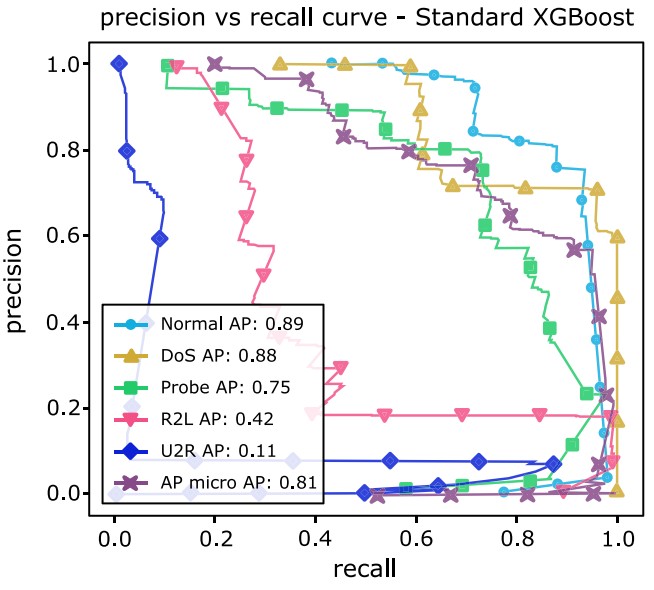

**Figure 5 PR curve of the basic XGBoost.**

NSL-KDD is imbalanced dataset, and the proposed CFAEE-SCA-XGBoost managed to achieve high performances (even for minority classes) for the accuracy and recall without modifying the original dataset. The PR curve of the basic XGBoost approach is shown in Fig. 5, while the PR curve of the proposed CFAEE-SCA-XGBoost method is presented in Fig. 6. To help visualising the difference and the improvements of the CFAEE-SCA-XGBoost method against the basic XGBoost, Fig. 7 depicts the precision *vs* recall curve comparison between the proposed CFAEE-SCA method and the basic XGBoost implementation. Finally, Table 14 presents the values of XGBoost parameters determined by the proposed CFAEE-SCA method.

## Experiments with UNSW-NB15 dataset

In the second set of experiments, the proposed model has been trained and tested by utilising the more recent UNSW-NB15 dataset, that was first proposed and analysed by *Moustafa & Slay (2015)* and *Moustafa & Slay (2016)*. The UNSW-NB15 dataset can be

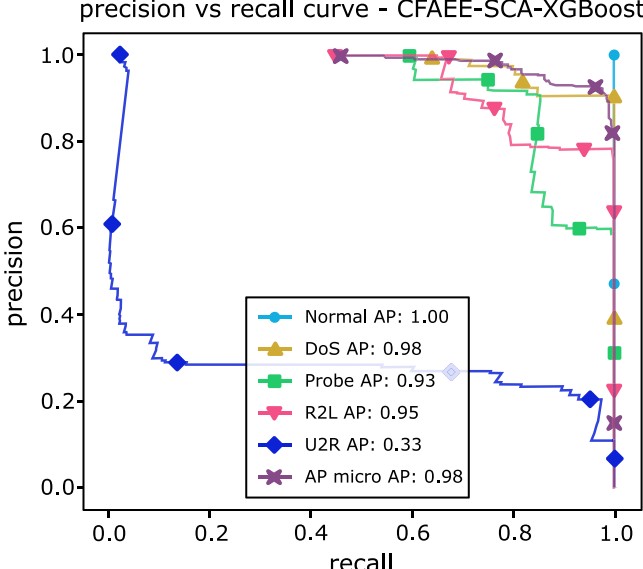

**Figure 6** PR curve of CFAEE-SCA-XGBoost.

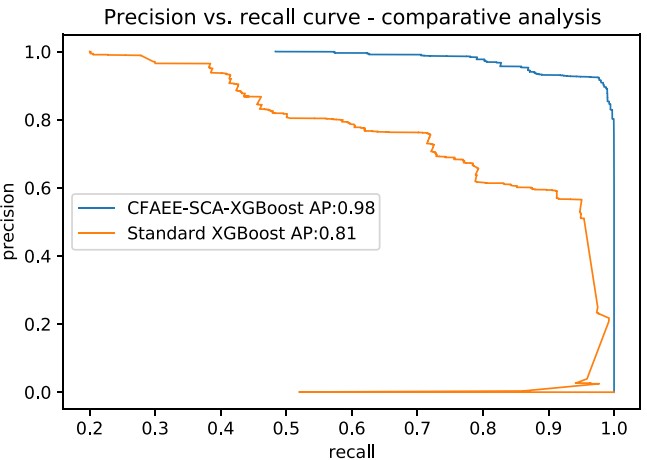

**Figure 7** PR curve comparative analysis between CFAEE-SCA-XGBoost and the basic XGBoost.

**Table 14** XGBoost parameter values after optimisation by CFAEE-SCA.

| Parameter | Determined value | Description |
|---|---|---|
| eta | 0.95 | Learning rate |
| max_depth | 3 | Max depth |
| min_child_weight | 1.74 | Min leaf weight |
| gamma | 0.1 | Related to loss function |
| sub-sample | 0.6 | Controls sampling to prevent over-fitting |
| colsample_bytree | 0.88 s | Controls feature sampling proportions |

**Table 15  Machine learning methods' parameter settings.**

| Method | Parameters |
| --- | --- |
| ANN | Adam solver, single hidden layer, *size* = {5, 10, 15, 30, 50, 100}, adaptive learning rate 0.02 |
| LR | random state set to 10, maximum 1,000 iterations |
| kNN | multiple models, *number_of_neighours* = {3, 5, 7, 9, 11} |
| SVM | regularisation parameter *C* = 1.12, *gamma* = 'scale', *kernel* = 'rbf' |
| DT | multiple models, *maximum_depth_value* = {2, 5, 7, 8, 9} |

retrieved from the following URL: https://github.com/naviprem/ids-deep-learning/blob/master/datasets/UNSW-NB15.md. Dataset features have been explained in *Moustafa & Slay (2015)*.

In total, the UNSW-NB15 dataset contains 42 features, out of which 39 are numerical, and three are categorical (non-numeric). The UNSW-NB15 contains two main datasets: UNSW-NB15-TRAIN, utilised for training various models and the UNSW-NB15-TEST, utilised for testing purposes of the trained models. The proposed model has been tested by following the instructions specified by *Kasongo & Sun (2020)*, in order to provide common grounds to compare the proposed model against their published results. The train set was divided into two parts, namely TRAIN-1 (75% of the training set) and VAL (25% of the training set), where the first part was used for training and the second part was used for validating before proceeding to test phase.

The UNSW-NB15 is comprised of instances belonging to the following categories that cover typical network attacks: Normal, Backdoor, Reconnaissance, Worms, Fuzzers, DoS, Generic, Analysis, Shellcode and Exploits. The research by *Kasongo & Sun (2020)* utilises XGBoost as the filter method for feature selection, and the features are normalised by using Min-Max scaling during the data processing. This was followed by application on various machine learning models, such as support vector machine (SVM), linear regression (LR), artificial neural network (ANN), decision tree (DT) and k-nearest neighbours (kNN).

The first phase of the experiments used the full feature size (total of 42 features) for the binary and multiclass configurations. The second part of the experiments utilised the feature selection powered by XGBoost as the filter method, resulting in the reduced number of features (19), that were subsequently used for the binary and multiclass configuration (details about the reduced features vector can be found in *Kasongo & Sun (2020)*). The parameters used for ANN, LR, kNN, SVM and DT are summarised in Table 15. It is important to state that the authors have implemented and recreated all experiments by utilising the same conditions as in *Kasongo & Sun (2020)* and tested them independently, with maximum *FFE* as termination condition.

The simulation results are shown in Tables 16–19. As mentioned before, the results for the ANN, LR, kNN, SVM and DT were obtained through independent testing by authors and those values have been reported and compared to the values obtained by the basic XGBoost (with default parameters' values), PSO-XGBoost, FA-XGBoost and the proposed CFAEE-SCA-XGBoost. The best result in each category is marked in bold text.

**Table 16 Comparative results of binary classification by utilising all 42 features.**

| Method | Acc training | Acc val | Acc test | Precision | Recall | F1-Score |
|---|---|---|---|---|---|---|
| ANN | 0.9448 | 0.9423 | 0.8670 | 0.8156 | 0.9803 | 0.8902 |
| LR | 0.9320 | 0.9286 | 0.7961 | 0.7331 | 0.9892 | 0.8424 |
| kNN | 0.9677 | 0.9357 | 0.8321 | 0.7916 | 0.9428 | 0.8603 |
| SVM | 0.7096 | 0.7062 | 0.6243 | 0.6089 | 0.8860 | 0.7117 |
| DT | 0.9366 | 0.9335 | 0.8811 | 0.8389 | 0.9648 | 0.9001 |
| XGBoost | 0.9526 | 0.9483 | 0.8712 | 0.8233 | 0.9824 | 0.8927 |
| PSO-XGBoost | 0.9713 | 0.9414 | 0.8914 | 0.8425 | 0.9894 | 0.9046 |
| FA-XGBoost | 0.9722 | 0.9427 | 0.8932 | 0.8457 | 0.9902 | 0.9061 |
| CFAEE-SCA-XGBoost | **0.9734** | **0.9469** | **0.8968** | **0.8493** | **0.9912** | **0.9103** |

Note:
The best achieved performance metric in all comparative analysis results tables are marked in bold.

**Table 17 Comparative results of binary classification by utilising 19 features.**

| Method | Acc training | Acc val | Acc test | Precision | Recall | F1-Score |
|---|---|---|---|---|---|---|
| ANN | 0.9377 | 0.9368 | 0.8441 | 0.7855 | 0.9852 | 0.8744 |
| LR | 0.8919 | 0.8924 | 0.7761 | 0.7316 | 0.9373 | 0.8218 |
| kNN | 0.9584 | 0.9471 | 0.8443 | 0.8028 | 0.9511 | 0.8709 |
| SVM | 0.7543 | 0.7553 | 0.6092 | 0.5893 | 0.9589 | 0.7299 |
| DT | 0.9413 | 0.9378 | 0.9086 | 0.8034 | 0.9841 | 0.8842 |
| XGBoost | 0.9516 | 0.9397 | 0.8478 | 0.7969 | 0.9788 | 0.8735 |
| PSO-XGBoost | 0.9599 | 0.9502 | 0.9121 | 0.8117 | 0.9859 | 0.8856 |
| FA-XGBoost | 0.9613 | 0.9514 | 0.9128 | 0.8134 | 0.9866 | 0.8873 |
| CFAEE-SCA-XGBoost | **0.9642** | **0.9539** | **0.9142** | **0.8167** | **0.9884** | **0.8891** |

Note:
The best achieved performance metric in all comparative analysis results tables are marked in bold.

**Table 18 Comparative results of multiclass classification by utilising all 42 features.**

| Method | Acc training | Acc val | Acc test | Precision | Recall | F1-Score |
|---|---|---|---|---|---|---|
| ANN | 0.7988 | 0.7957 | 0.7559 | 0.7991 | 0.7557 | 0.7655 |
| LR | 0.7552 | 0.7395 | 0.6556 | 0.7693 | 0.6547 | 0.6663 |
| kNN | 0.8174 | 0.7681 | 0.7012 | 0.7578 | 0.7018 | 0.7202 |
| SVM | 0.5345 | 0.5271 | 0.6113 | 0.4749 | 0.6201 | 0.5378 |
| DT | 0.7766 | 0.7735 | 0.6601 | 0.7977 | 0.6604 | 0.5109 |
| XGBoost | 0.8155 | 0.7868 | 0.7395 | 0.7981 | 0.7264 | 0.7609 |
| PSO-XGBoost | 0.8216 | 0.7985 | 0.7592 | 0.8013 | 0.7611 | 0.7683 |
| FA-XGBoost | 0.8233 | 0.8007 | 0.7604 | 0.8028 | 0.7626 | 0.7698 |
| CFAEE-SCA-XGBoost | **0.8247** | **0.8029** | **0.7630** | **0.8045** | **0.7654** | **0.7724** |

Note:
The best achieved performance metric in all comparative analysis results tables are marked in bold.

Table 16 reports the findings of the experiments with different ML approaches, basic XGBoost and three XGBoost metaheuristics models for the binary classification that utilises the complete feature set of the UNSW-NB15 dataset. On the other hand, Table 17

**Table 19 Comparative results of multiclass classification by utilising 19 features.**

| Method | Acc training | Acc val | Acc test | Precision | Recall | F1-Score |
|---|---|---|---|---|---|---|
| ANN | 0.7944 | 0.7890 | 0.7748 | 0.7949 | 0.7751 | 0.7725 |
| LR | 0.7252 | 0.7179 | 0.6527 | 0.7085 | 0.6526 | 0.6594 |
| kNN | 0.8267 | 0.7989 | 0.7232 | 0.7726 | 0.7232 | 0.7385 |
| SVM | 0.5358 | 0.5295 | 0.6151 | 0.5392 | 0.6150 | 0.5127 |
| DT | 0.7876 | 0.7845 | 0.6759 | 0.7967 | 0.6758 | 0.6927 |
| XGBoost | 0.7987 | 0.7903 | 0.7592 | 0.7931 | 0.7429 | 0.7528 |
| PSO-XGBoost | 0.8324 | 0.8016 | 0.7765 | 0.7993 | 0.7772 | 0.7756 |
| FA-XGBoost | 0.8347 | 0.8033 | 0.7784 | 0.8015 | 0.7796 | 0.7789 |
| CFAEE-SCA-XGBoost | **0.8378** | **0.8069** | **0.7803** | **0.8046** | **0.8015** | **0.7824** |

**Note:**
The best achieved performance metric in all comparative analysis results tables are marked in bold.

depicts the results of the binary classification over the reduced feature set of the UNSW-NB15 dataset.

Tables 18 and 19 present the results obtained by different ML models, basic XGBoost and three XGBoost metaheuristics models for the multiclass classification that uses the complete and reduced feature vectors, respectively. In every table, Acc training represents the accuracy obtained over the training data, Acc val stands for the accuracy obtained over the validation data partition, and finally, Ac test denotes the accuracy obtained over the test data.

The experimental findings over the USNW-NB15 IDS dataset clearly indicate the superiority of the hybrid swarm intelligence and XGBoost methods over the standard machine learning approaches. All three XGBoost variants that use metaheuristics significantly outperformed all other models, both in case of binary classification and in case of multiclass classification. Similarly, the swarm based approaches outperformed the traditional methods for both complete feature set, and for the reduced number of features. Among the three XGBoost variants that use metaheuristics for optimisation, the PSO-XGBoost achieved the third place, basic FA-XGBoost finished second, while the proposed CFAEE-SCA-XGBoost obtained the best scores on all four test scenarios by the significant margin. This conclusion further establishes the proposed CFAEE-SCA-XGBoost method as a very promising option for the intrusion detection problem.

## CONCLUSIONS

This article has presented a proposed an improved FA optimisation algorithm CFAEE-SCA, that was devised with a goal to overcome the deficiencies of the basic FA metaheuristics. Several modifications have been made to the basic algorithm, including explicit exploration mechanism, gBest CLS strategy, and hybridisation with SCA to further enhance the search process. The proposed improved metaheuristics was later used to optimise the XGBoost classifier for the intrusion detection problem. The CFAEE-SCA-XGBoost framework has been proposed, based on the XGBoost classifier, with its hyper-parameters, optimised and tuned using the newly proposed CFAEE-SCA algorithm. The proposed model was trained and tested for network intrusion detection using two well-

known datasets: NSL-KDD and UNSW-NB15 dataset. The proposed model, supported by the CFAEE-SCA algorithm, outperformed the variation supported by the original FA algorithm, the PSO-XGBoost and the basic implementation of the XGBoost, that were used in the comparative analysis.

The experimental results show that the CFAEE-SCA-XGBoost model obtained the best accuracy compared to the original model and suggest the potential for using swarm intelligence algorithms for NIDS. These results uncover possible future areas for research and application.

### Funding
The authors received no funding for this work.

### Competing Interests
The authors declare that they have no competing interests.

### Author Contributions
- Miodrag Zivkovic conceived and designed the experiments, performed the computation work, prepared figures and/or tables, and approved the final draft.
- Milan Tair conceived and designed the experiments, performed the computation work, prepared figures and/or tables, and approved the final draft.
- Venkatachalam K. conceived and designed the experiments, performed the computation work, prepared figures and/or tables, and approved the final draft.
- Nebojsa Bacanin performed the experiments, analyzed the data, authored or reviewed drafts of the paper, and approved the final draft.
- Štěpán Hubálovský performed the experiments, analyzed the data, authored or reviewed drafts of the paper, and approved the final draft.
- Pavel Trojovský performed the experiments, analyzed the data, authored or reviewed drafts of the paper, and approved the final draft.

### Data Availability
The data in the Supplememental File is available from the NSL-KDD dataset (public dataset for intrusion detection): https://www.unb.ca/cic/datasets/nsl.html.

### Supplemental Information
Supplemental information for this article can be found online at http://dx.doi.org/10.7717/peerj-cs.956#supplemental-information.

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
