# Peer review of "Novel hybrid firefly algorithm: an application to enhance XGBoost tuning for intrusion detection classification"

_PeerJ Computer Science, doi:10.7717/peerj-cs.956_

## Round 0.1 · original submission · Major Revisions

A major revision is necessary before further processing. Please provide a detailed response letter along with your revision. Thanks.

Reviewer 1 ·

Basic reporting

1. In the abstract of this paper, the authors consider that “One of the greatest issues from the domain of network intrusion detection systems are relatively high true positives and false negatives rates.” Nevertheless, a high true-positive rate is precisely what researchers in the IDS need and pursue. So why is it an issue to address? Please clarify whether the authors are a clerical error. If not, please fully explain. If it is indeed a clerical error, please check the article carefully to avoid ambiguity and misunderstanding caused by similar problems.

2. In the introduction of this paper, the authors need to verify their understanding of the basic concepts. In particular, the significance of false-positive rate and false-negative rate in IDS. If these two definitions are not proper, are subsequent performance metrics based on these defined correctly? Are subsequent experimental designs and data reliable?

Experimental design

1. The authors have carried on the sufficient experiment, the multi-level multi-stage comparison, and submitted the source code. However, could the authors consider presenting experimental data more diversely? Large and unlabeled forms can make reading difficult. In addition, the dataset is somewhat monolithic so that authors can try multiple datasets rather than just NSL-KDD.

Validity of the findings

1. Reducing false positive and false negative rates is an urgent problem in IDS. However, the authors need to explain more fully (1) why XGBoost is used as a classifier for IDS to solve this problem (especially when the model has been plagued by a high false-positive rate and false-negative rate) and (2) why the hyperparameters of XGBoost are optimized by a heuristic algorithm rather than other methods. It requires not only theoretical analysis but also experiments.

2. Please further describe the challenges met and problems solved in the research. And match them to the contribution of this work.

3. In the contribution of the paper, the authors state that “A novel enhanced FA metaheuristics has been developed by specifically targeting the well-known deficiencies of the original FA implementation;” Please specify the “well-known weaknesses” and how to overcome them? Given such shortcomings, why is it still necessary to base FA metaheuristics design solution? Why can’t other similar schemes replace FA metaheuristics?

4. The authors provide a clear and sufficient description of improvements of the firefly algorithm but lack the content of the overall scheme. Please describe the complete process of this scheme in an appropriate way.

Annotated reviews are not available for download in order to protect the identity of reviewers who chose to remain anonymous.

Reviewer 2 ·

Basic reporting

Basic reporting is average. Language and presentation should be improved. There are several typos that should be addressed. Authors need to make a careful revision of the document in this regard.
Further IDS using ML and DL models is used widely discussed in the literature, including XGBoost algorithm and also with other ML based techniques. Further, there are techniques based on XGBoost to take care of encrypted environment. For eg.
Intrusion detection model using fusion of chi-square feature selection and multi class SVM, 2017.
Anomaly Detection Using XGBoost Ensemble of Deep Neural Network Models, 2021.
Under this backdrop authors need to provide a justification for the new proposal.
Is the focus is to improve FA or improve IDS accuracy? The statements in second page are confusing.
Contributions listed do not indicate the IDS aspect. So the understanding goes is that the authors have worked on the optimization algorithm and in order to validate it they have used IDS datasets. In such case, the title of the paper is not justifiable.
There are several computational aspects in the proposed algorithm which do not have specific way of handling. For e.g Perform search operations. What is the complexity?
What is K in the algorithm? First of all,, what are the inputs to the algorithm and what are the output?
It is better if the extensive literature reported are summarized in the background section.

Experimental design

Results are fine. However there are other datasets such as UNSW, VIT Sparc, etc.

Validity of the findings

Results are validated using the standard dataset. Hence they are validated.

Additional comments

NIL

·

Basic reporting

The English is clear.

Experimental design

A design setup of the proposed architecture is required for analyzing the whole approach.

Validity of the findings

What is the motivation for this work as there are many other models developed for IDS integrated with meta heuristic algorithms?

The proposed model is tested on NSL-KDD dataset which is very old. The authors should select recent public benchmark intrusion detection datasets and show the superiority of the proposed model.

Though ensemble models result in better accuracy, their execution time is greater in comparison to base classifiers. The authors should analyze the computational complexity of the proposed work.

A comparative analysis of the proposed model with state-of-the-art intrusion detection models should be performed.

What is the effect of optimizing the XGBoost parameters in the proposed approach. The authors should analyze in detail how these parameters had a positive impact in performance in discussion section.

Additional comments

Certain intermediate results of CEC2013 can be separately added in a appendix section at the end of the paper.

---

## Round 0.2 · Minor Revisions

The paper is potentially publishable. Please make a revision according to the comments. Thanks.

Reviewer 2 ·

Basic reporting

It is improved from the previous version.

Experimental design

While the design looks good, experimental results are not presented clearly. Some of the tables are not visible.

Validity of the findings

Yes.

Additional comments

While answering the questions/suggestions some of the responses are overlapping towards the queries. The same answer is provided for different queries. This confusion should be clarified.

·

Basic reporting

Clear

Experimental design

All comments are addressed by the authors

Validity of the findings

All comments are addressed by the authors

Additional comments

-

---

## Round 0.3 · accepted · Accept

The paper can be accepted. Congratulations.

Reviewer 2 ·

Basic reporting

Basic reporting is fine.

Experimental design

Experimental design is explained fine.

Validity of the findings

It is experimentally validated by the authors.

Additional comments

Article is improved from the current version.